Manuscript prepared for Geosci. Model Dev.
with version 2015/04/24 7.83 Copernicus papers of the LaTeX class copernicus.cls.
Date: 27 May 2016

# The co-condensation of semi-volatile organics into multiple aerosol particle modes

Matthew Crooks[1], Paul Connolly[1], David Topping[1], and Gordon McFiggans[1]

[1]The School of Earth, Atmospheric and Environmental Science, The University of Manchester, Oxford Road, Manchester, M13 9PL

*Correspondence to:* matthew.crooks@manchester.ac.uk

**Abstract.** An existing equilibrium partitioning model for calculating the equilibrium gas/particle concentrations of multiple semi-volatile organics within a bulk aerosol is extended to allow for multiple involatile aerosol modes of different sizes and chemical compositions. In the bulk aerosol problem the partitioning coefficient determines the fraction of the total concentration of semi-volatile material that is in the condensed phase on the aerosol. This work modifies this definition for multiple polydisperse aerosol modes to account for multiple condensed concentrations; one for each semi-volatile on each involatile aerosol mode. The pivotal assumption in this work is that each aerosol mode contains an involatile constituent thus overcoming the potential problem of smaller particles evaporating completely and then condensing on the larger particles to create a monodisperse aerosol at equilibrium. The resulting coupled non-linear system is approximated by a simpler set of equations in which the organic mole fraction in the partitioning coefficient is set to be the same across all modes. By perturbing the condensed masses about this approximate solution a correction term is derived which accounts for much of the removed complexities. This method offers a greatly increased efficiency in calculating the solution without significant loss in accuracy, thus making it suitable for inclusion in large scale models.

## 1 Introduction

Volatile and semi-volatile compounds are of key importance to a wide variety of industries (Biniecka and Caroli (2011)) including atmospheric science (Topping et al. (2013), Topping and McFiggans (2012)), pharmaceuticals (Sitaramaraju et al. (2008), Wang et al. (2011)), food and drink (Vernocchi et al. (2008), de Roos (2003), Hui et al. (2010), Mendes et al. (2012)), water treatment (Zarra et al. (2009)) and perfume (Morris (1984)). Many of these industries exploit the scented properties of a large number of semi-volatiles and an understanding of their behaviour is crucial in order to identify, produce and capture certain aromas as well as controlling the rate of their subsequent evaporation in, for example, the perfume industry. In the water treatment sector the odorous properties are not considered a favourable quality and inhibiting their release is important. The production of volatile compounds from all of these industries, however, can have far reaching negative consequences on

both health and the environment (Klein (1995), Epstein and Gibson (2013), Wu et al. (2009)) as well as influencing weather and climate (Morgan et al. (2010), Morris (1984)).

This paper approaches the study of semi-volatile compounds from an atmospheric and cloud physics perspective but the equilibrium partitioning theory presented herein is applicable to the wide variety of problems outlined above. The condensed phases of the semi-volatile compounds in the atmosphere occur within aerosol particles which can have a profound effect on cloud properties and climate. The ability of an aerosol particle to act as a cloud condensation nucleus (CCN) is determined by many factors including size, number concentration and chemical composition, as well as the supersaturation of the air (Pruppacher and Klett (1977)). An increase in number concentration of a monodisperse aerosol which acts as a CCN will, in general, produce a greater number of smaller cloud droplets as there are more particles competing for the same available water (Twomey (1959)). In contrast, according to Köhler theory, the larger aerosol particles in a polydisperse system will have a greater affinity to activate into cloud droplets and their presence can deplete the available water vapour more quickly, due to their quicker growth rates. Consequently, this can suppress the supersaturation which prevents the smaller particles from activating (Ghan et al. (1998)).

The aforementioned effects of aerosol size and number concentration can alter the precipitation rate and, as a result, the lifetime of clouds (Stevens and Feingold (2009), Albrecht (1989)). In addition to a direct effect of an increase in number of cloud droplets leading to both an increase in reflected shortwave radiation and absorbed longwave radiation (McCormick and Ludwig (1967), Chýlek and Coakley Jr (1974)) there is a complicated interdependency between cloud longevity and albedo (Twomey (1974), Twomey (1977)). The result is an approximate $-0.7$ W m$^{-2}$ increase in mean global radiative forcing; although this is subject to a large degree of uncertainty, which is on the order of the total radiative forcing from anthropogenic activity (IPCC (2013), Lohmann et al. (2000)).

Quantifying the size and chemical composition of individual aerosol particles within a population is complicated due to them being a heterogeneous mix of primary and secondary aerosol. Primary aerosol particles are emitted directly from biogenic and anthropogenic sources. Although some models exist to simulate purely semi-volatile primary particles (Tsimpidi et al. (2014)) in this paper they are assumed to contain at least a small portion of non-volatile material. Secondary aerosol forms by condensation of gases onto the particles. This process is fairly well understood for inorganic gases (Hallquist et al. (2009)), however, there is significant uncertainty associated with the formation of secondary organic aerosol (SOA) from volatile organic compounds (VOCs). Part of this uncertainty is a result of the poorly quantified process of oxidation of VOCs to produce SOA and other VOCs with reduced volatility (Jimenez and et. al. (2009), Yu (2011)).

Over remote continental regions between $5\%$ and $90\%$ of the total aerosol mass can be made up of organic material (Andreae and Crutzen (1997), Zhang et. al. (2007), Gray et al. (1986)) and a significant proportion of this can be from secondary sources. SOA in such large quantities will act to

increase the size of the particles as well as significantly changing their chemical composition. The effect of the increased size, and consequently soluble mass, is found to be the dominant effect on cloud; increasing the number of cloud droplets and subsequently decreasing the critical supersaturation (Dusek et al. (2006), Topping et al. (2013)).

It is estimated that $10^4 - 10^5$ organic species have been measured in the atmosphere but this may only be a small proportion of the total (Goldstein and Galbally (2007)). Of this number fewer than 3000 have actually been identified (Simpson et al. (2012), Borbon et al. (2013)). It is, therefore, impractical to try to model each compound individually and two popular methods have been developed to simulate multiple organic species using equilibrium absorptive partitioning theory (Pankow (1994)); a method of calculating the equilibrium condensed concentrations without solving the computationally expensive dynamic condensation and evaporation processes. The first noteworthy approach (Odum et al. (1996)) proposes an empirically fitted relation derived from two-compound experiments which benefits from its simplicity but, as with any empirical relation, has possibly limited applicability outside of the original constraints. In particular, these models are found to be unrealistically sensitive to changes in concentration of the organics (Cappa and Jimenez (2010)). The second (Donahue et al. (2006)) uses a volatility basis set; binning large numbers of semi-volatile organic compounds (SVOCs) into a small number of representative species with effective saturation concentrations. This method was later extended to account for the co-condensation of both water and SVOCs (Barley et al. (2009)) by treating the water as an additional volatile substance.

A limitation of equilibrium absorptive partitioning theory (Pankow (1994)) is the assumption of well mixed particles. Although some experimental results indicate that aerosol particles containing SOA can exist in highly viscous states (Vaden et al. (2011), Cappa and Wilson (2011)), it is still an active area of research especially in relation to high relative humidities and the effects on cloud droplet formation. Numerical models investigating the effect of diffusion within aerosol particles (Zobrist et al. (2011), Smith et al. (2003)) indicate a liquid phase of the particles above above 50% relative humidity, which is typically of atmospheric relevance.

The focus of this paper is on the condensation of SVOCs within the atmosphere and we assume that there are sufficient aerosol particles that the process of condensation onto existing particles is the dominant sink for the SVOCs over nucleation of new particles. The existing particles, which act as a seed for the condensation, are assumed to be involatile although in many applications particles that have previously nucleated from oxidation of extremely low volatility compounds can also be approximated as being involatile (Ehn et al. (2014)). Consequently, every particle considered in this paper is assumed to have at least a small portion of an involatile compound which cannot evaporate into the vapour phase. This assumption is crucial in calculating the equilibrium vapour/condensed phases on multiple modes as without it a polydisperse equilibrium does not exist. In the purely volatile particle case the smaller particles evaporate completely thus allowing the larger particles to take up additional condensed mass to produce a monodisperse aerosol at equilibrium. We present a

new formulation for calculating this equilibrium condensed phase across multiple aerosol modes to be used as a computationally efficient approximation to the dynamic condensation process.

In this paper we present a model for equilibrium absorptive partitioning of water and SVOCs onto multiple involatile aerosol modes using the volatility basis set method of Donahue et al. (2006). All compounds are considered to be at least partially soluble so that the involatile aerosol influences the partitioning of the SVOCs and the particles are assumed to be always well mixed. The concentration of a SVOC in the condensed phase on one mode has the effect of reducing the total concentration available to the other modes and as a result the equilibrium vapour/condensed phases of all the organics on all the involatile modes must be solved for simultaneously. By assuming all modes have approximately the same organic mole fraction a method of solution is derived which has a high degree of accuracy while being significantly more efficient than a standard numerical algorithm would be. Condensed concentrations of organics are compared against equilibrium calculated from a modified parcel model and are found be comparable.

## 2    Equilibrium Absorptive Partitioning Theory for Size Independent Bulk Aerosol

Before introducing the new multiple mode equilibrium partitioning theory, we first review existing partitioning theories, beginning with that for a bulk aerosol, (Donahue et al. (2006)). Multiple organic species are binned according to their saturation concentration, $\mathcal{C}_j^*$, using a $\log_{10}$ volatility basis set with values from $1 \times 10^{-6} \mu\mathrm{g}\ \mathrm{m}^{-3}$ to $1 \times 10^3 \mu\mathrm{g}\ \mathrm{m}^{-3}$ (Cappa and Jimenez (2010), Donahue et al. (2006)). We use the calligraphic notation, $\mathcal{C}_j^*$, for the saturation concentration measured in $\mu\mathrm{g}\ \mathrm{m}^{-3}$ to distinguish from the analogous molar quantity (Barley et al. (2009)), $C^*$, in units of $\mu\ \mathrm{mol}\ \mathrm{m}^{-3}$ obtained by dividing $\mathcal{C}_j^*$ by the molecular weight. The value of $C_j^*$ is proportional to the saturation vapour pressure, $p_j^o$ (atm), of the organics in the $j^{th}$ bin at temerature, $T_0$, through the equation

$$C_j^* = \frac{p_j^o \gamma_j 10^6}{RT_0}. \tag{1}$$

Here $\gamma_j$ is the the activity coefficient of the organics in the $j^{th}$ bin, $R$ is the universal gas constant $(\mathrm{m}^3\ \mathrm{atm}\ \mathrm{mol}^{-1} \mathrm{K}^{-1})$ and $T_0$ is the temperature (K). The saturation vapour pressure is dependent on the temperature, $T_0$, which is taken to be 298.15K. Conversion to a $C^*$ at another temperature, $T$, can be done using the Clausius Clapeyron equation,

$$C^*(T) = C^*(T_0)\frac{T_0}{T}exp\left[-\frac{\Delta H_{vap}}{R}\left(\frac{1}{T} - \frac{1}{T_0}\right)\right],$$

where $\Delta H_{vap}$ is the enthalpy of vapourisation of the organic compounds and is taken to be 150 kJ $\mathrm{mol}^{-1}$ in the current work.

Water and other inorganic semi-volatile compounds can additionally be considered by binning them along with the organics. For simplicity we ignore inorganic semi-volatiles and for the purposes of applying the theory to the particular problem of cloud physics later in the paper we treat water as a seperate volatile compound so that its abundance can be varied independently of the organics.

The total concentration of the semi-volatile in the $j^{th}$ volatility bin is defined by $C_j$ ($\mu$ mol m$^{-3}$) and is decomposed into a vapour and a condensed phase,

$$C_j = C_j^v + C_j^c, \tag{2}$$

indicated by the superposed $v$ and $c$ respectively. The total concentration of all compounds in the condensed phase, $C_{OA}$, is the sum of the concentration of ions of involatile aerosol, $C^o$, the condensed water, $C^w$, and each organic component in the condensed phase,

$$C_{OA} = C^o + C^w + \sum_k C_k^c. \tag{3}$$

The condensed water can be calculated assuming ideality by equating the saturation ratio, $S$, to the mole fraction of water,

$$S = \frac{C^w}{C^o + C^w + \sum_k C_k^c}. \tag{4}$$

Equation (4) can be rearranged to make $C^w$ the subject and then substituted into (3) and factorised to give

$$C_{OA} = \frac{1}{1-S}\left(C^o + \sum_k C_k^c\right). \tag{5}$$

The analogous expression to (4) for the organics is

$$\frac{C_j^v}{C_j^*} = \frac{C_j^c}{C_{OA}}, \tag{6}$$

where the saturation ratio has been replaced by the ratio of the vapour pressure to the saturation vapour pressure of the $j^{th}$ component using (1).

Eliminating $C_j^v$ from (2) using (6) and rearranging gives the condensed concentration of the $j^{th}$ organic component in terms of the total concentration

$$C_j^c = C_j \left(1 + \frac{C_j^*}{C_{OA}}\right)^{-1}. \tag{7}$$

The partitioning coefficient is defined as

$$\xi_j = \left(1 + \frac{C_j^*}{C_{OA}}\right)^{-1}, \tag{8}$$

so that $C_j^c = \xi_j C_j$. Since $C_{OA}$ depends on the $C_j^c$ this problem must be solved iteratively; calculating the $C_j^c$ given some $C_{OA}$, using these values to update the value of $C_{OA}$ and repeating until equation (5) is satisfied within some tolerance.

## 3 Equilibrium Absorptive Partitioning Theory for a Monodisperse Aerosol

The theory in the previous section is independent of the size of the aerosol particles. We extend this theory to include size dependence for a monodisperse aerosol. The partitioning of water is influenced

by the size of the aerosol through the Kelvin factor,

$$K^w = \exp\left(\frac{4M_w\sigma}{\rho_w RTd}\right),$$

where $M_w$, $\rho_w$ and $\sigma$ are the molecular weight, density and surface tension of water, and $d$ is the diameter of the wet aerosol particles. The Kelvin factor multiplies the right-hand side of equation (4), see, for example, Rogers and Yau (1996),

$$S = \frac{C^w K^w}{C^o + C^w + \sum_k C_k^c}. \tag{9}$$

Consequently, the right hand side of equation (5) becomes

$$C_{OA} = \frac{K^w}{K^w - S}\left(C^o + \sum_k C_k^c\right), \tag{10}$$

and for notational simplicity we define the prefactor on the right-hand side by $\eta$,

$$\eta = \frac{K^w}{K^w - S}.$$

The Kelvin factor for the organics, $K_j$, is defined by Cai (2005) as

$$K_j = \exp\left(\frac{4M_j\sigma}{\rho_j RTd}\right), \tag{11}$$

where $M_j$ and $\rho_j$ are the molecular weight and density of the organics in the $j^{th}$ volatility bin, and $\sigma$ is the surface tension of the particle with condensed organics and water. This is included into equation (6) in an analogous way to the Kelvin factor for water to give

$$\frac{C_j^v}{C_j^*} = \frac{C_j^c K_j}{C_{OA}}, \tag{12}$$

and consequently the partitioning coefficient is defined as

$$\xi_j = \left(1 + \frac{C_j^* K_j}{C_{OA}}\right)^{-1}. \tag{13}$$

This set of algebraic equations again needs to be solved iteratively as in the previous section but with the addition of calculating the diameter, and consequently Kelvin term, from the $C_j^c$ at each iteration.

## 4 Equilibrium Absorptive Partitioning Theory for a Conglomeration of Particles of Multiple Sizes and Composition

An aerosol population commonly comprises particles of different sizes and chemical composition. In this section we consider aerosol which can be decomposed into multiple monodisperse aerosol modes. We propose a novel extension to the theory of the previous sections which is applicable to these more diverse conglomerations of particles and therefore suitable for application to a wider class of problems.

The most significant change to the theory for multiple modes is that the equations must explicitly take into account that a single molecule of an organic cannot condense onto more than one mode. Consider the 10 organic molecules and 2 involatile particles shown in blue and green respectively in Figure 1. Suppose the subset highlighted in pink are known to condense onto the left involatile particle; these molecules cannot also condense onto the right particle. Hence, the only the molecules

available to condense onto the right particle are those circled by the dashed line. The converse is also true; if the molecules highlighted in pink in Figure 2 are known to condense onto the right particle then only the remaining molecules are free to condense onto the left particle. As such, the concentration of organics in the $j^{th}$ volatility bin which is available to the $i^{th}$ mode is the total concentration in the $j^{th}$ volatility bin minus the condensed concentrations of the organics in the $j^{th}$

bin on the other modes. It is this quantity that is multiplied by the partitioning coefficient in the multiple mode case.

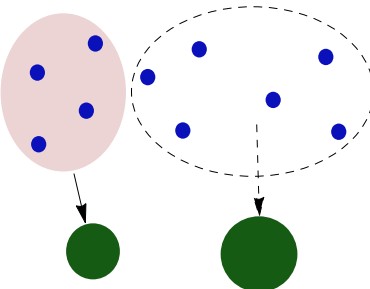

**Figure 1.** A representation of the condensation of a SVOC, shown by the blue dots, onto two non-volatile particles, shown by the green circles. The SVOC is divided into two groups; those that condense onto the left involatile particle and those remaining to equilibrate on the right particle.

equilibrate

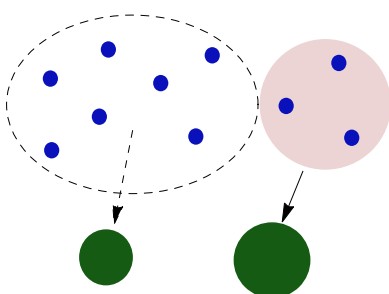

**Figure 2.** Same as Figure 1 but with the two non-volatile particles swapped.

We make the following changes to extend the theory in the previous section to the multiple mode case:

– The condensed mass of the $j^{th}$ organic component will now be split across several modes. Define the condensed concentration of the $j^{th}$ organic species on the $i^{th}$ involatile mode by $C_{ij}^c$ so that the condensed phase $C_j^c$ can we expressed as the sum

$$C_j^c = \sum_r C_{rj}^c.$$ (14)

For consistency, a subscript $i$ refers to the involatile mode and $j$ refers to the $j^{th}$ volatility bin. Dummy indices $r$ and $k$ are used in the summations to make clear which terms are being summed and are restricted to sums over $i$ and $j$, respectively. Subsequently, the total concentration of semi-volatiles in the $j^{th}$ volatility bin is decomposed into a single vapour phase and the sum of the multiple condensed phases

$$C_j = C_j^v + \sum_r C_{rj}^c.$$ (15)

The total concentration in the condensed phase on the $i^{th}$ mode is given by the sum of the concentrations of the involatile constituent and all of the condensed organics and water on the $i^{th}$ mode,

$$C_{OA,i} = C_i^o + C_i^w + \sum_k C_{ik}^c,$$

which can equally be expressed in an analogous way to (10) as

$$C_{OA,i} = \frac{K_i^w}{K_i^w - S} \left( C_i^o + \sum_k C_{ik}^c \right).$$ (16)

Here

$$K_i^w = \exp\left( \frac{4M_w \sigma}{\rho_w R T d_i} \right),$$

is the Kelvin factor of water and $C_i^o$ is the number of $\mu$ mol per cubic meter of the $i^{th}$ involatile aerosol mode. $d_i$ is the diameter of the $i^{th}$ mode with both condensed water and organics. For notational simplicity we define

$$\eta_i = \frac{K_i^w}{K_i^w - S},$$ (17)

so that (16) can be written as

$$C_{OA,i} = \eta_i \left( C_i^o + \sum_k C_{ik}^c \right).$$ (18)

– The partitioning coefficient, given by (13), depends on the material properties and size of the involatile aerosol particles, through $C_{OA}$ and $K_j$, and consequently each volatility bin can have a different coefficient for each mode. The Kelvin factor for the $j^{th}$ volatility bin, (11), is modified to depend on the diameter of the $i^{th}$ composite mode, $d_i$,

$$K_{ij} = \exp\left(\frac{4M_j\sigma}{\rho_j RT d_i}\right),$$

Subsequently, the multiple mode analogy to (12) on the $i^{th}$ mode is

$$\frac{C_j^v}{C_j^*} = \frac{C_{ij}^c K_{ij}}{C_{OA,i}}, \tag{19}$$

This can be used to eliminate the vapour concentration from (15)

$$C_j = \frac{C_j^* C_{ij}^c K_{ij}}{C_{OA,i}} + \sum_r C_{rj}^c.$$

The concentration $C_{ij}^c$ can be extracted from the sum on the right-hand side and the equation rearranged to give

$$C_j - \sum_{r \neq i} C_{rj}^c = C_{ij}^c \left(1 + \frac{C_j^* K_{ij}}{C_{OA,i}}\right). \tag{20}$$

We denote the partitioning coefficient for the $j^{th}$ organic component onto the $i^{th}$ involatile mode by $\xi_{ij}$ and define it in an analogous way to (13),

$$\xi_{ij} = \left(1 + \frac{C_j^* K_{ij}}{C_{OA,i}}\right)^{-1}. \tag{21}$$

– The total concentration of the $j^{th}$ organic which is available to the $i^{th}$ involatile mode, as previously discussed, can now be defined

$$C_{ij} = C_j - \sum_{r \neq i} C_{rj}^c, \tag{}$$

or alternatively

$$C_{ij} = C_j - \sum_r C_{rj}^c + C_{ij}^c. \tag{22}$$

– An expression for the condensed concentration of the $j^{th}$ organic on the $i^{th}$ involatile mode can now be expressed as the product of the partitioning coefficient with $C_{ij}$,

$$C_{ij}^c = \frac{C_j - \sum_r C_{rj}^c + C_{ij}^c}{1 + \frac{C_j^* K_{ij}}{C_{OA,i}}}, \tag{23}$$

which is a rearrangement of (20). Replacing $C_{OA,i}$ using (18) yields

$$C_{ij}^c = \frac{C_j - \sum\limits_r C_{rj}^c + C_{ij}^c}{1 + \dfrac{C_j^* K_{ij}/\eta_i}{C_i^o + \sum\limits_k C_{ik}^c}}. \tag{24}$$

The non-linear nature of these equations poses a problem for obtaining a solution. Numerical methods exist for solving such problems but require an initial guess for the solution. Furthermore, there are multiple unknowns and multi-dimensional non-linear solvers are too computationally expensive to implement in a global climate model. In the remainder of the paper we present a method for obtaining an approximate solution to equations (24) which facilitates a numerical solver in computing the full solution. This approximation, however, is sufficiently accurate that an analytic perturbation method, derived in Section 6, can be used to reduce the inaccuracies and render the costly numerical solution redundant.

## 5 Leading order solution

### 5.1 Motivation

Low accuracy solutions to equations (24) for the equilibrium concentration in the condensed phase can be found using a brute force trial and error method. 50 such solutions for between 2 and 5 involatile modes are shown in Figure 3 with each calculated using different randomised inputs for the parameters taken from Table 1. The concentrations of organics in each volatility bin are kept in the proportions shown in Table 2 but are rescaled by a random factor of between 0 and 100. Plotted in the graph is the total organic mole fraction which we define as the concentration of semi-volatile material on a mode divided by the total number of ions in the aerosol. For the $i^{th}$ mode this is

$$\theta_i = \frac{\sum\limits_k C_{ik}^c}{C_i^o + \sum\limits_k C_{ik}^c}, \tag{25}$$

Each solution is represented by several dots joined by a vertical line with the $y$ coordinate of the dots showing the mole fraction of one of the modes. These are plotted against the average mole fraction of all the modes in that solution on the $x$ axis. As can be seen from Figure 3, even though the mole fractions are different, they never deviate too far from the mean, shown by the dashed line. Assuming the same mole fraction for all the modes allows approximate values of $C_{OA,i}$ to be found in terms of a single parameter. This greatly simplifies the problem and can provide a reasonable guess for the non-linear solver.

We emphasise that although the mole fractions of the different modes are comparable, the equilibrium partitioning solution is still dependent on the individual material properties of the involatile

aerosol with number concentration, size, molecular weight, density and van't Hoff factors all influencing the number of moles of the involatile compound and consequently the number of moles of condensed organics required to give the appropriate mole fraction.

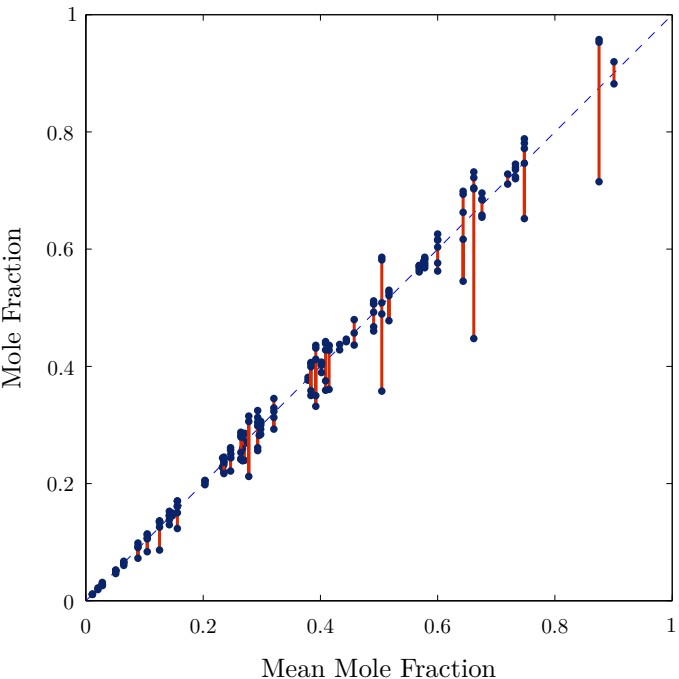

**Figure 3.** The mole fractions of multiple different involatile aerosol modes plotted against mean mole fraction on the $x$ axis. 50 solutions are shown with parameters randomly chosen from the ranges shown in Table 1 with $R_H = 0\%$.

**Table 1.** Randomised variables used to plot Figure 3. $n_i$, $d_i$, $\rho_i^o$ and $M_i^o$ are the number concentration, diameter, density and molecular weight of the $i^{th}$ involatile aerosol, respectively. $\nu_i^o$ and $\nu_j$ are the van't Hoff factors of the involatile constituents and organics.

| variable | $n_i$ (pcc) | $d_i$ (nm) | $\rho_i^o$ (kg m$^{-3}$) | $\rho_i$ (kg m$^{-3}$) | $M_j, M_i^o$ (kg mol$^{-1}$) | $\nu_i^o$ | $\nu_j$ |
|---|---|---|---|---|---|---|---|
| min | 50 | 50 | 500 | 500 | 0.1 | 1 | 0 |
| max | 500 | 500 | 2000 | 1500 | 0.4 | 3 | 1 |

**Table 2.** The distribution of total concentrations of organics used in the simulations.

| $\log_{10} \mathcal{C}^*$ | -6 | -5 | -4 | -3 | -2 | -1 | 0 | 1 | 2 | 3 |
|---|---|---|---|---|---|---|---|---|---|---|
| $C_j (\mu$ mol m$^{-3}$) | 0.025 | 0.05 | 0.1 | 0.15 | 0.30 | 0.40 | 0.8 | 1.5 | 2.1 | 5 |

## 5.2 Derivation of a Solution with an Average Mass Fraction

The solution derived under the assumption of a common mole fraction, $\theta$, is referred to as the leading order solution and is denoted, $\bar{C}_{ij}^c$. Rearranging (25) with $\theta_i$ replaced by $\theta$ gives

$$\frac{\theta}{1-\theta} = \frac{\sum\limits_k \bar{C}_{ik}^c}{C_i^o}.$$

The left-hand side of this equation is the same for all the modes and consequently the right-hand side must also be the same. For notational simplicity we combine the left-hand side into a single parameter, $\beta$,

$$\beta = \frac{\theta}{1-\theta} = \frac{\sum\limits_k \bar{C}_{ik}^c}{C_i^o}, \tag{26}$$

and subsequently

$$\sum\limits_k \bar{C}_{ik}^c = \beta C_i^o. \tag{27}$$

The wet diameter of the combined aerosol is assumed to be sufficiently large that the Kelvin factors for the SVOCs can be approximated by 1. The same is not true for the Kelvin factors for water, $K_i^w$. If these are assumed to take the value 1 then the definition of $\eta_i$, as given by (17), will lead to a gross over estimate of the condensed water close to cloud base, becoming infinite as the relative humidity becomes 100%.

The governing equations for the leading order solution are obtained by making the approximation given by (27) together with setting $K_{ij} = 1$,

$$\bar{C}_{ij}^c = \frac{C_j - \sum\limits_r \bar{C}_{rj}^c + \bar{C}_{ij}^c}{1 + \dfrac{C_j^*/\eta_i}{C_i^o(1+\beta)}}. \tag{28}$$

The interdependence of the $\bar{C}_{ij}^c$ can be eliminated to obtain an explicit expression for each in terms of the mole fraction $\beta$,

$$\bar{C}_{ij}^c = \frac{C_i^o(1+\beta)\left(1 - \phi_j(\beta, \eta_i)\right)C_j}{C_j^*/\eta_i}, \tag{29}$$

the algebra of this step is given in A. The dependence of $\phi_j$, given by (A2), is expressed explicitly for clarity. Both sets of equations are equivalent to one another and either can be used to find the $\bar{C}_{ij}^c$. The coupled nature of equations (28), however, requires a matrix inversion to solve and this will take more time to calculate than the solution to equations (29).

The complexity of the summation of $\bar{C}_{ij}^c$ in the denominator has now been removed in favour of a single parameter, $\beta$. This can be solved for using a much quicker one-dimensional root find algorithm which iterates the value of $\beta$ and solves the linear system of equations at each step to find $\bar{C}_{ij}^c$. Equations (28) together with solving equation (26) for each value of $i$ produces an over

determined system for $\bar{C}_{ij}^c$ and $\beta$ with more equations than unknowns. To overcome this we only solve (26) for $i = 1$.

While the dependence of $\eta_i$ on $\bar{C}_{ij}^c$ has not been removed it is sufficiently weak that the values of $\bar{C}_{ij}^c$ calculated at the previous iteration of $\beta$ can be used to evaluate the Kelvin factor for water and therefore $\eta_i$; simultaneously converging on the correct value as $\beta$ is found.

### 5.3 Results

The leading order solution given in the previous section is calculated and used as an initial guess for solver of the full non-linear problem; a comparison of the two solutions are presented here. In order to test the theory over a large parameter space values of the inputs are chosen from Table 1 and the models are run for between 2 and 6 modes. Solutions from only one mode are plotted for each run to avoid a bias towards the solutions for 6 mode runs which would otherwise have three times as many points plotted as the 2 mode runs.

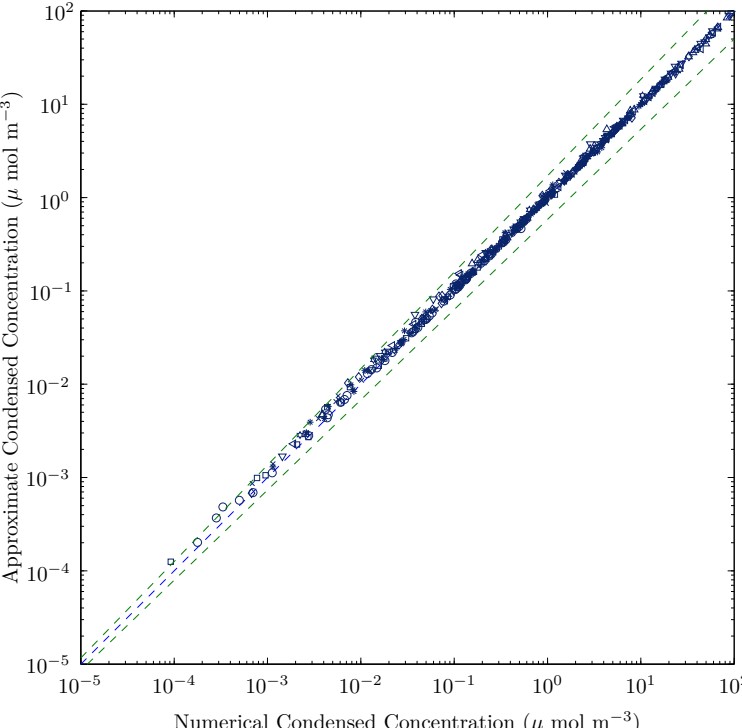

**Figure 4.** Comparison of the condensed concentrations of organics from the leading order solution against that calculated using the non-linear solver for randomly chosen parameters from Table 1. The solution is calculated for between 2 and 6 modes and only the condensed concentrations on the first mode are plotted. The different volatility bins are distinguished by the shape of the points. Equality between the solutions is shown by the dashed blue line and the green lines shown where the approximation over or under estimates by a factor of 2.

Figure 4 shows the individual condensed masses of the 10 volatility bins, each with different shaped points. The blue dashed line shows equality between the approximation and the full solution and the green dashed lines show 50% inaccuracies in the approximations. The very strong correlation between the full non-linear solution and the leading order solution demonstrates that the assumption of a common mole fraction made in the previous section not only offers an efficient way of calculating an initial guess for the non-linear solver but actually offers an accurate approximation to the solution for the condensed masses of the organics.

Figure 5 compares the organic mass fractions, defined as the total mass of condensed organics divided by the total mass of the aerosol, from the full non-linear solution and the leading order solution from the different runs. A range of relative humidities and van't Hoff factors of the involatile aerosol are used; these values are shown above each plot. Small values of the van't Hoff factor appear to have a significant effect on the mass fraction as the correlation of the right-hand three plots is worse. This is caused by the need to divide by the van't Hoff factor when converting from number of ions to mass. The result is a high sensitivity of the mass to slight inaccuracies in the number of ions for van't Hoff factors close to zero; this is addressed in Section 6.

Similarly, lower values of the relative humidity reduce the correlation of the two solutions with almost perfect agreement in the lower two plots for $R_H = 99.999\%$. This is likely a result of higher relative humidities increasing the wet diameter of the droplets and subsequently reducing the Kelvin factor closer to 1. Since the leading order solution sets the Kelvin factors of the organics to 1, higher values of $R_H$ improve this approximation. At $R_H = 99.999\%$ there increased water contend of the wet aerosol will further act to increase the value of $C_{OA}$ and subsequently lead to a significantly higher proportion of the organics being in the condensed phase. With nearly all the organics in the condensed phase there is much less potential for errors in the approximation.

## 6   Perturbation Solution

Although the condensed masses computed using the approximation derived in Section 5 did not completely agree with the full non-linear solution, the errors were small relative to the size of the $C_{ij}$. We propose a correction term to this approximation which improves the accuracy and also takes into account the Kelvin terms. Suppose the actual condensed masses, $C_{ij}^c$, can be obtained from the leading order solution, $\bar{C}_{ij}^c$, by adding a small perturbation

$$C_{ij}^c = \bar{C}_{ij}^c + \hat{C}_{ij}^c,$$

where we assume $|\hat{C}_{ij}^c| \ll |\bar{C}_{ij}^c|$. The Kelvin factors also depend on the final condensed mass and for clarity ought to be written $K_{ij} = K_{ij}(C_{ij}^c)$. This, however, adds much complexity. Therefore, it is assumed that the leading order solution provides a suitably accurate approximation to the condensed masses of semi-volatiles for the purposes of calculating the Kelvin factors. Consequently, we make the additional approximation $K_{ij}(C_{ij}^c) \approx K_{ij}(\bar{C}_{ij}^c)$ which we denote $\bar{K}_{ij}$. Similarly, $K^w(C_{ij}^c) \approx$

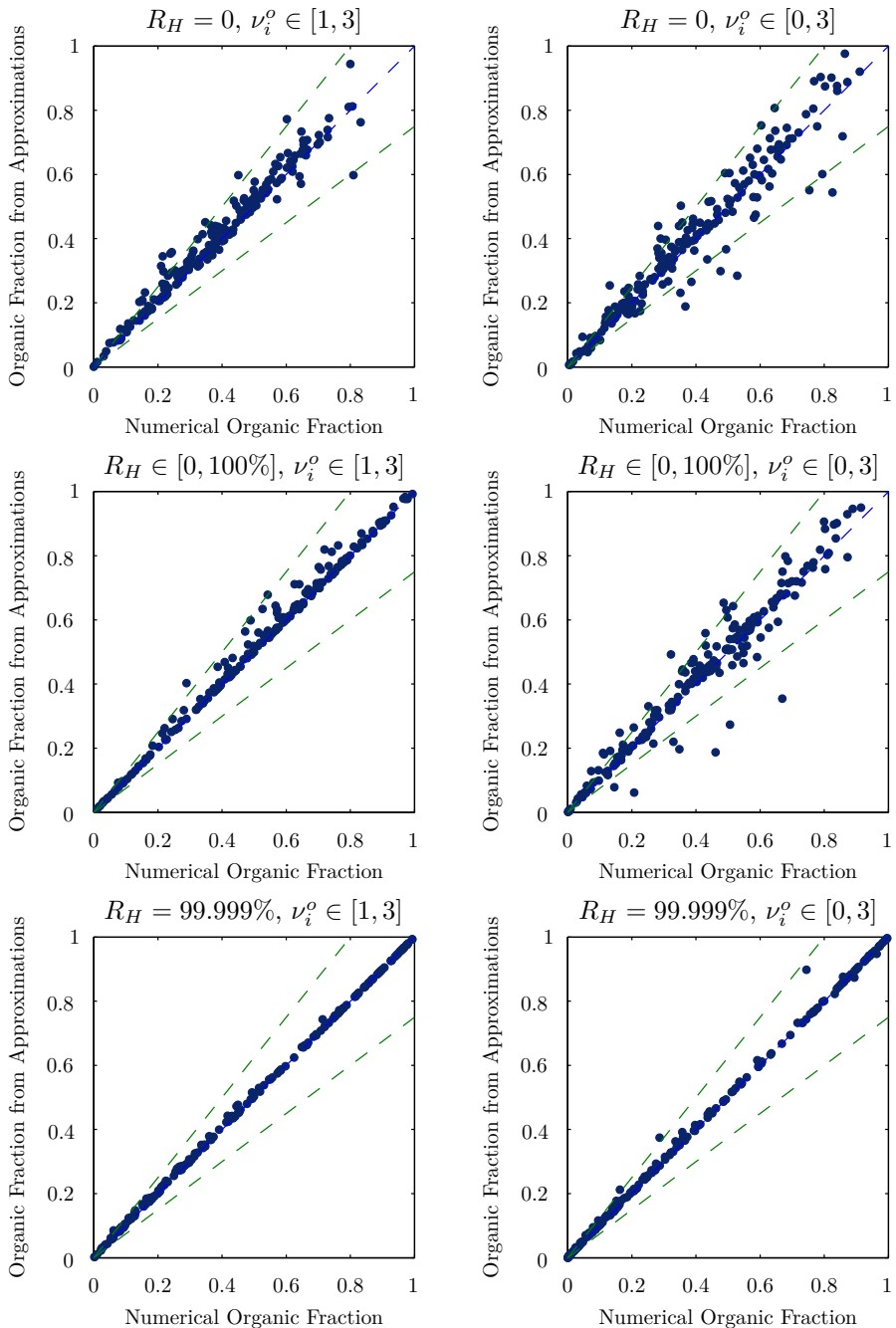

**Figure 5.** Comparison of the organic mass fraction from the two approximations against that calculated using the non-linear solver for randomly chosen parameters from Table 1. The solution is calculated for between 2 and 6 modes and only the mass fraction of the first is plotted. 25% error margins are shown by the dashed green lines.

$K^w(\bar{C}_{ij}^c)$ and consequently $\eta_i(C_{ij}^c) \approx \eta_i(\bar{C}_{ij}^c) = \bar{\eta}_i$. We substitute the perturbation into the equations

(24) together with these approximations to give

$$\bar{C}_{ij}^c + \hat{C}_{ij}^c = \frac{C_j - \sum_r \bar{C}_{rj}^c + \bar{C}_{ij}^c - \sum_r \hat{C}_{rj}^c + \hat{C}_{ij}^c}{1 + \dfrac{C_j^* \bar{K}_{ij}/\bar{\eta}_i}{C_i^o + \sum_k \bar{C}_{ik}^c + \sum_k \hat{C}_{ik}^c}}. \tag{30}$$

Assuming the perturbed quantities, $\hat{C}_{ij}^c$, are small we can linearise these equations to give

$$\left(1 - \bar{\xi}_{ij}\right)\hat{C}_{ij}^c + \bar{\xi}_{ij}\sum_r \hat{C}_{rk}^c - \mathcal{L}_{ij}\sum_k \hat{C}_{ij}^c = \frac{C_j - \sum_r \bar{C}_{rj}^c + \bar{C}_{ij}^c}{1 + \dfrac{C_j^* \bar{K}_{ij}/\bar{\eta}_i}{C_i^o + \sum_k \bar{C}_{ik}^c}} - \bar{C}_{ij}^c. \tag{31}$$

the algebra for which is given in B. The $\bar{\xi}_{ij}$ are the partitioning coefficients, (21), calculated using
the leading order solution; explicit expressions are given by (B2). The $\mathcal{L}_{ij}$ are defined in B and are coefficients that depend on the $\bar{C}_{ij}^c$ as well as the parameters of the problem but are independent of the perturbations. Therefore, these equations are linear in the perturbations and can be solved much quicker than the full set of non-linear equations.

   If the leading order solution, $\bar{C}_{ij}^c$, were the exact answer then the right-hand side would be simple
a rearrangement of the equations (24) and would be zero. In such a situation, the left-hand side would also have to be zero resulting in a homogeneous system of coupled linear equations with only the trivial solution in which $\hat{C}_{ij}^c = 0$. However, the leading order solution does not satisfy the full system of equations but is "close" by some measure. Consequently, the right-hand side of (31) is "small" and defines the size of the small perturbation $\hat{C}_{ij}^c$. The additional correction term can be calculated
by a simple matrix inversion owing to the linear nature of equations (31) which, as we shall see in the following section, produces a much more accurate solution in a fraction of the time it would take to run the non-linear solver.

### 6.1 Results

Figure 6 is a replica of Figure 4 with the addition of the condensed concentrations calculated using the first order correction term shown in red. The perturbation solution can be seen to offer an improved approximation to the leading order solution, especially above $0.1\,\mu$ mol m$^{-3}$ where the agreement with the non-linear solution is almost perfect. At lower concentrations the correction term reduces the degree to which the leading order approximation over predicts the condensed concentra-
tions compared to the non-linear solution.

   The organic mass fractions calculated from the perturbation solution are plotted against those calculated using the non-linear solver in Figure 7 and are shown in red. The graphs show a marked improvement on the leading order solution, which are shown in navy, across all parameter space.

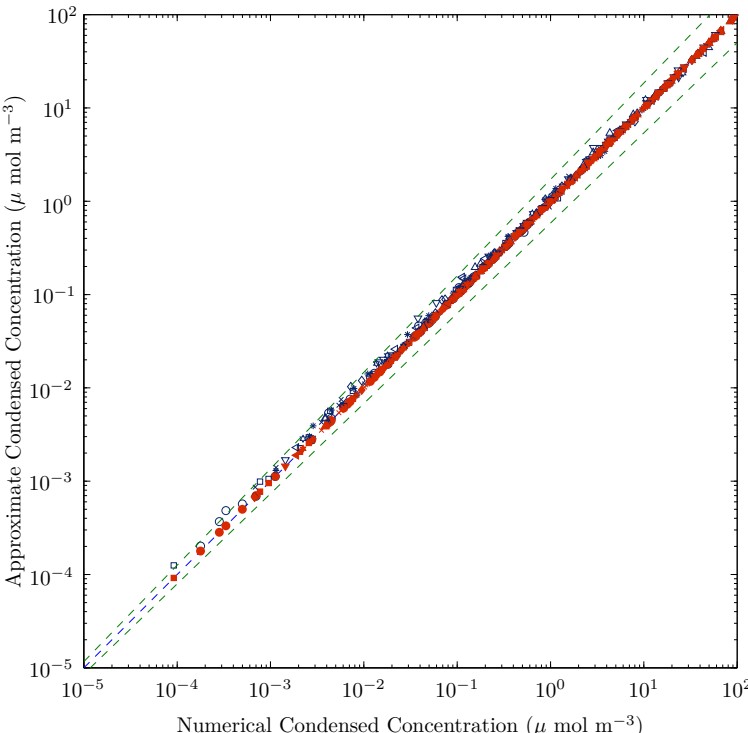

**Figure 6.** Same as Figure 4 with the addition of the perturbation solution shown in red.

Adding the first order correction term reduces the errors to less than 20% in the worst cases of the
top right plot compared to nearly 50% in the leading order solution. The general trends of worse
correlation for low $R_H$ and $\nu_i^o$ observed in Figure 5 are replicated in the perturbation solution but to
a much lesser extent. This is due to the inclusion of a leading order Kelvin factor for the organics that
reduces the influence of the $R_H$ and improvements in the accuracy of the condensed concentrations
reduce the subsequent errors in the masses.

Figure 8 shows the percentage errors in the calculated mass fractions when compared to the solu-
tion from the non-linear solver. The top two plots, which had the worst correlation in Figure 7, show
that the leading order solution produces errors in the organic mass fraction of below 20% in nearly
all cases and these reduce to 10% when the correction term in the perturbation solution is included.
The affect of van't Hoff factors close to zero is most notable in the middle two plots where the errors
increase about 3 fold compared to when the van't Hoff factors are greater than 1; middle right and
left plots, respectively. The errors in the perturbation solution rarely exceed 10% across the entire
parameter space and are only a few percent at cloud base, shown by the lower 2 plots.

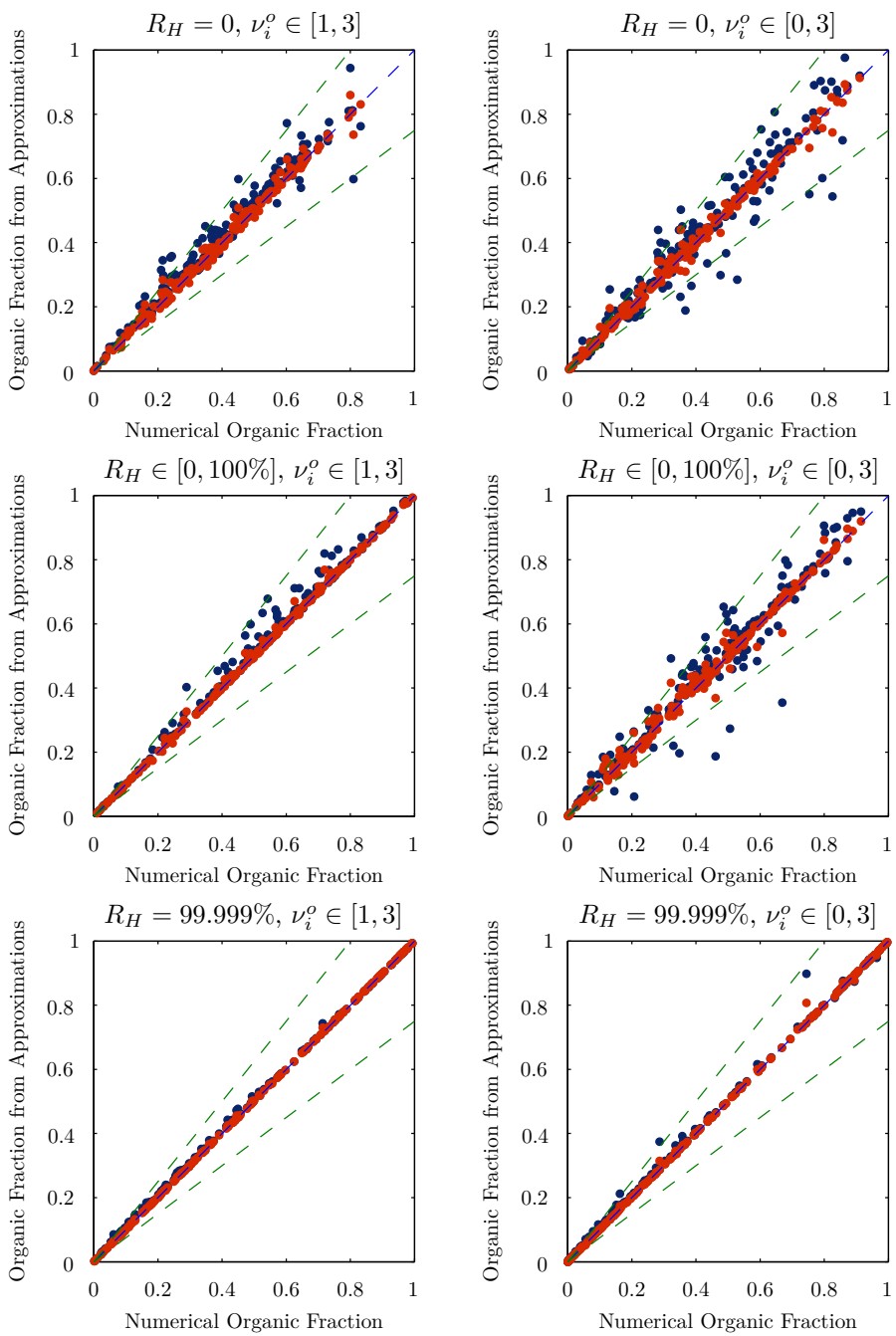

**Figure 7.** Same as Figure 5 with the addition of the perturbation solution shown in red.

## 7 Comparison Between the Partitioning Theory and Parcel Model Simulations

The partitioning theory presented in the preceding sections calculates the equilibrium of each organic
between the vapour and the condensed phases. If a dynamic model of condensation of organics is

left to run for long enough it ought to equilibrate on the same solution. We test this here for two modes as verification of the extension of the partitioning theory to multiple modes.

A dynamic parcel model by Topping et al. (2013) is modified to maintain constant temperature, pressure and water vapour mass; the former two set to 293.15 K and 95000 Pa respectively and the latter is calculated from the specified relative humidity. In the simulations the time taken for small particles to grow to equilibrium is on the order of hundreds of seconds whereas larger particles take thousands. The large particles, however, should take up more of the condensed concentration of the organics due to their higher value of $C_{OA}$. As a consequence, the vapour phase equilibrates with the total condensed concentration very quickly but places too large a proportion on the smaller mode and insufficient on the larger. Due to the bulk system reaching equilibrium the large particles can only grow at the same rate that the small particles shrink and this process can take millions of seconds of simulation time to correct. We derive a more accurate initial condition by allowing time for the large particles to equilibrate with the vapour before placing the small particles into the model. This reduces the amount of organic vapour available to condense onto the smaller mode and prevents the gross over calculation of the condensed concentration on this mode in the initial time period.

**Table 3.** Material parameters used in the parcel model simulations.

| variable | $\rho_i^o$ (kg m$^{-3}$) | $\rho_j$ (kg m$^{-3}$) | $M_j$ (kg mol$^{-1}$) | $M_i^o$ (kg mol$^{-1}$) | $\nu_i^o$ | $\nu_j$ |
|---|---|---|---|---|---|---|
| value | 1770 | 1500 | 0.132 | 0.2 | 3 | 1 |

**Table 4.** Range of values assigned to the number concentration, $n_i$, and involatile particle diameter, $d_i$, used in the parcel model simulations.

| variable | $n_i$ (pcc) | $d_i$ (nm) |
|---|---|---|
| mode 1 | 100-1000 | 25-100 |
| mode 2 | 50 | 125 |

The parcel model was run twice with the parameters given in Tables 3 and 4. The first mode had a number concentration of 1000 pcc and a diameter of 50 nm in the first simulation and 100 nm in the second. The second mode was given initial periods of 300 and 10 seconds of simulation time, respectively, before the smaller mode was added. This speeds up the equilibration period to allow a better comparison with the equilibrium partitioning theory. Subsequent time evolution of the condensed concentrations are shown by the solid lines in Figure 9 for a relative humidity of 50%. Bulk equilibrium is reached by about 1000 seconds with the individual modes reaching equilibrium by approximately 20000 seconds. The parcel model converges on a value which is in excellent agreement with the partitioning theory shown by the dashed lines.

Figure 10 compares the condensed concentrations for a broader range of values of the diameter of

the first mode; values are denoted above each plot. The top of the red sections of each bar depict the

total condensed concentration in each volatility bin from the parcel model and the equivalent quantity

from the partitioning theory is shown by the black crosses; the two agree almost exactly. The height

of the green bar shows the condensed concentrations on the first mode from the parcel model and,

as expected, an increase in diameter of these particles results in both an increase in total condensed

concentration and the proportion of the which is on the first mode. The small particles of diameter

25 nm in the top left plot only account for a small proportion of the condensed concentration, as

shown by the small area of green compared to the large area of red. When the size of these particles

is increased to 100 nm they take up almost all of the organic material in the condensed phase;

the lower right bars have a much larger area of green than red. The analogous quantities from the

partitioning theory are shown by the dashed yellow lines which coincide with the tops of the green

bars to a high degree of accuracy. The condensed concentrations on the second mode are shown

by the height of the red sections of the bars which, when measured from the $x$ axis, are shown by

the horizontal black dashed lines across each bar. These too agree exceptionally well with the pink

dashed lines which are the equivalent values from the partitioning theory.

Similar plots are shown in Figure 11 but this time the size of the first mode is fixed at 50 nm and

the number concentration is varied instead; the values used are shown above each plot. As previously

seen, the equilibrium partitioning theory agrees incredibly well with the parcel model simulations

across the parameter space explored. Interestingly in these plots the total condensed concentration

does not increase significantly when the number concentration of the first mode is increased from

100 pcc to 750 pcc. The proportion on each of the two modes, however, does increase, with almost

equal amounts placed on each of the two modes when the first mode has a number concentration of

750 pcc.

### 7.1   Application to Atmospherically Relevant Problems

Aerosol in the atmosphere is often characterised by multiple polydisperse modes with particles in

each mode composed of a single chemical composition. The sizes of particles in each mode are

described by size distributions which are a convenient way of treating a polydisperse aerosol as each

size distribution can be considered as a single entity.

A common assumption is that the diameters of the particles, $d$, are lognormally distributed so that

the number of aerosol particles, $n$, per logarithmic size interval is given by

$$\frac{dn}{d\ln d} = \frac{n_i}{\sqrt{2\pi}\ln\sigma}\exp\left[-\left(\frac{\ln\left(\frac{d}{d_m}\right)}{\sqrt{2}\ln\sigma}\right)^2\right].$$

Here $N$ is the total number of particles and $d_m$ and $\ln\sigma$ are the median diameter and geometric

standard deviation. Multiple lognormal size distributions can be added together to form more diverse

aerosols

$$\frac{dn}{d\ln d} = \sum_i \frac{n_i}{\sqrt{2\pi}\ln\sigma_i} \exp\left[-\left(\frac{\ln\left(\frac{d}{d_{m,i}}\right)}{\sqrt{2}\ln\sigma_i}\right)^2\right],$$

where $d_{m,i}$ and $\ln\sigma_i$ are the median diameter and geometric standard deviation of the $i^{th}$ mode.

The difficulty with applying equilibrium partitioning theory to size distributions is that particles of different sizes have different Kelvin factors and the semi-volatiles will consequently partition non-uniformly across the particles. We replace the Kelvin factors by effective values calculated using the median diameters of each size distribution. In Section 7.3.2 we show how two different methods can be used to approximate the size distribution in two different regimes.

**Table 5.** Parameter values used in the first set of parcel model simulations

| variable | $n$ (pcc) | $d_m$ (nm) | $\ln\sigma$ |
|----------|-----------|------------|-------------|
| mode 1   | 200       | 25         | 0.5         |
| mode 2   | 50        | 125        | 0.1         |

We begin by comparing the limiting behaviour of the parcel model against the equilibrium partitioning theory, presented in this paper, when each mode is represented by lognormal size distributions. To speed up the equilibration period in the parcel model we allow the large particles an initial period to interact with the SVOC vapours before adding the small particles to the simulation.

Section 7.3 shows the results of the parcel model for time periods of up to 3 days when all particles are allowed to interact with the SVOC vapours from the beginning of the simulation. Estimates of the residence times of aerosol particles in the atmosphere vary significantly with as long as 65 days being reported in one study (Marenco and Fontan (1973)). The majority of the literature, however, predicts residence times of 5 - 15 days (Paspastefanou (2006), Ahmed et al. (2004), Balkanski et al. (1993)) and so the 3 day time is chosen as this is well within the lifetime of aerosol particles in the atmosphere.

### 7.2 Equilibrium Comparisons

The parcel model is run with the parameters given in Table 5 together with the material properties from Table 3 and the second mode is given an initial period of 1500 seconds of simulation time before the smaller mode is added. Again, this speeds up the convergence of the parcel model towards equilibrium to allow comparison between its limiting behaviour and our equilibrium partitioning theory applied to lognormal modes. The subsequent time evolution of the condensed concentrations are shown by the solid lines in Figure 12 for a relative humidity of $0\%$ and $90\%$. Bulk equilibrium is reached by about 1000 seconds and the parcel model converges on a value which is in good agreement with the partitioning theory shown by the dashed lines.

Figure 13 shows the condensed concentrations from Figure 12 on the two individual modes with the left plots showing the smaller mode and right plots showing the larger. Simulations are run for 20000 and 40000 seconds for $R_H = 0\%$ and $R_H = 90\%$, respectively, and although the organic shown by the yellow line has not completely reached equilibrium on the first mode it is assumed to be sufficiently close. The condensed concentrations of organics on the larger mode from the partitioning theory are in good agreement with the equilibrium in the parcel model, however, they over predict the concentration on the smaller mode by a factor of two. Due to the total condensed concentrations being in good agreement, an under estimate on the second mode must be accompanied by an over estimate of equal magnitude on the first mode. This then results in a more significant relative error on the first mode because the condensed concentrations are an order of magnitude smaller.

The equilibrium calculated using the parcel model is further compared against the partitioning theory in Figure 14 for a range of relative humidities. The stacked bar charts show the proportions of organics in each volatility bin which are in the vapour phase and the condensed phases on each mode. Total condensed concentrations from the partitioning theory are shown by the crosses which lie almost exactly at the top of the red sections which mark the analogous quantity from the parcel model at equilibrium. As previously discussed, the partitioning theory under predicts the condensed concentrations on the first mode (dashed yellow line) compared to the parcel model (green bar) and the converse is true for the larger mode (dashed pink line and dashed black lines). The effect of the increased relative humidity is to increase the total condensed concentration of organics; all of the organics in the five lowest volatility bins are in the condensed phase even at $R_H = 0\%$ and so this extra organic must come from the higher volatility bins. This is seen by the larger red and green regions on the higher volatility bins.

The proposed reason for the discrepancies in the condensed masses on the smaller mode is the effect of the Kelvin factor. This is more variable for smaller particles and so using an effective value for all particles within a lognormal mode results in errors. To demonstrate this theory we have carried out further simulations with a smaller mode with median diameter of 50 nm. These are shown in Figure 15 for relative humidities of 0% and 90%. As can be seen, the errors in condensed mass on the smaller mode are eradicated and the two equilibrium solutions are in perfect agreement. We conclude, therefore, that equilibrium partitioning can very accurately calculate the equilibrium condensed mass on lognormal modes if the median diameter is above about 50 nm. In some situations it may be deemed sufficiently accurate to use equilibrium partitioning with lognormal size distributions at smaller diameters, especially when off-set against the reduction in computational complexity compared to solving the dynamic condensation process.

## 7.3   3 day Simulation Comparisons

The previous section compared how accurately the new model predicts the condensed masses of SVOCs in multiple modes when each mode is represented by a lognormal size distribution. We

now consider under what time scales it can be assumed that equilibrium partitioning is in sufficient agreement with the dynamic condensation.

### 7.3.1 Condensed Mass

Figure 16 shows the condensed mass on the two lognormal modes from the parcel model at a range of times up to three days. These are compared against the solution to equilibrium partitioning. The smaller mode has a median diameter of 25 nm and the relative humidity is kept at a constant 90%. The equilibrium plot is the same as in Figure 14 and as previously discussed, the parcel model solution is about 30% higher than the equilibrium partitioning solution. Compared against the dynamical solution, the equilibrium partitioning theory performs less well on the smaller mode. Even by 12 hours, the lower 5 volatility bins are significantly higher in the dynamic solution than at equilibrium. This is because the lower $C^*$ value means these compounds evaporate and condense more slowly than the higher volatility compounds. The additional condensed mass in these volatility bins causes additional mass in the higher volatility bins to remain in the condensed phase on the smaller mode. By 72 hours the two solutions are in better agreement but there is still a large disparity in the lower 4 volatility bins.

Figure 17 shows the relative error of the condensed mass from equilibrium partitioning against the dynamic solution. At equilibrium the total errors in mass are about 35% with the majority of this mass coming from the higher volatility bins; shown by the more abundant purple shades. After 12 hours of dynamic condensation the errors are about 70% and are much more equally apportioned across all the volatility bins. About half of this error, however, is attributed to the use of a lognormal size distribution in the equilibrium partitioning theory. Even after three days there is still over 60% errors in the partitioning theory solution.

In the previous section it was found that using a smaller mode with a median diameter of 50 nm rather than 25 nm gave much better agreement between the equilibrium partitioning theory and the parcel model at equilibrium. The parcel model additionally converges on equilibrium much quicker in the 50 nm case. Figure 18 shows the condensed mass from the parcel model at a range of times after initiation and even after just 2 hours the solution is comparable to equilibrium partitioning with almost perfect agreement in the upper four volatility bins. By 24 hours the errors reduce even further and are restricted to the lower 5 volatility bins. Figure 19 shows the relative errors in the equilibrium partitioning solution for each of the plots in Figure 18. At equilibrium the errors are less than 4% so using a lognormal size distribution with median diameter of 50 nm resolves the issues seen in the 25 nm case. In the 2 hour plot the total error in mass assuming equilibrium partitioning is only about 17% on the smaller mode and this decreases to 12% by 24 hours.

When studying longer term aerosol transport it is possible that equilibrium partitioning theory may provide a sufficiently accurate and significantly less computationally expensive method of cal-

culating condensed masses of SVOCs compared to direct numerical calculation of the dynamic condensation process.

### 7.3.2 Size Distribution

Calculating the change in size distribution as a result of condensing SVOCs is very much a dynamical process (Pandis et al. (1993), Wexler and Seinfeld (1990), Zhang et al. (2012)), not only are the organic compounds condensing onto each mode as a whole but they are also evaporating and condensing between the different sizes of particles within each mode. We compare the size distribution from the parcel model against two methods of approximating this size using equilibrium partitioning theory that represent two different regimes.

The first, as suggested by Connolly et al. (2014), approximates a very rapid condensation process such as that experienced near cloud as a result of rapid increases in relative humidity. This method is applied to the dry aerosol size distribution (without the associated water) and maintains a constant arithmetic standard deviation and decreases the geometric standard deviation in order to maintain mass within the system. Over longer periods of time the SVOCs on small particles evaporate and condense onto larger particles producing a broadening of the size distribution returning the geometric standard deviation to a value more similar to its initial value. This is the second method employed.

Figures 20 and 21 show the size distributions from the parcel model against the two methods applied to equilibrium partitioning. The constant arithmetic standard deviation is shown by the dashed line and the solid lines show the result of only changing the median diameter. In most cases, the wet size distribution is reasonably well approximated with a constant geometric standard deviation. The exception being the earlier plots for the 25 nm mode and is a result of the SVOCs not being close to equilibrium. In many situations, wet aerosol sizes are an important factor, such as in radiative forcing calculations, and equilibrium partitioning together with maintaining a constant geometric standard deviation could provide a quick method of accurately capturing this size distribution.

More interesting is the dry aerosol case where only the aerosol constituent of the particles is plotted. At equilibrium in Figure 21 the size distribution from the parcel model is accurately captured by the equilibrium partitioning theory together with the constant geometric standard deviation approximation. In the 2 hour solution, however, the size distributions are more accurately approximated by the dashed line showing the constant arithmetic standard deviation method. This is especially evident in the smaller mode which is more actively still undergoing condensation. This is in agreement with the initial proposal put forward in Connolly et al. (2014). By 24 hours, however, both modes have broadened and parcel model has become more similar to the constant geometric standard deviation model.

## 8 Conclusions

This paper presents both a model and an efficient and accurate method of solution which is suitable for investigations in a wide range of research areas that use equilibrium partitioning theory. Of particular interest to the authors is cloud droplet activation parameterisations and, ultimately, inclusion in global climate models.

The model itself predicts the equilibrium condensed concentrations of organics onto multiple monodisperse aerosol modes incredibly accurately compared to the equilibrium solution from a dynamic parcel model. This holds true for a range of particle sizes, number concentrations and relative humidities.

The condensed mass calculated using equilibrium partitioning theory with lognormal size distributions of involatile particles agrees very well with the limiting behaviour of the parcel model if the median diameter of the modes is above 50 nm. Below this size the Kelvin term becomes important and results in non-negligible errors. Additionally, under dynamic conditions the dynamic solution with a median diameter of 50nm reaches a state that is close to equilibrium after only 2 hours and becomes increasingly more accurate with time.

Calculating the size distribution of lognormal modes is a dynamical process, even after the condensed mass reaches equilibrium. Two methods are suggested to calculate the size distribution in two different regimes. The wet aerosol size distribution is accurately predicted from equilibrium partitioning theory assuming constant geometric standard deviation from just 2 hours after initiation of the parcel model. In comparison, the dry aerosol size distribution passes through a phase in which the size distribution narrows during a rapid condensation phase and then broadens out over a longer period of time.

Changes in condensed mass as a result of relative humidity tend to only have a significant effect on the higher volatile compounds that can equilibrate quickly. In the atmosphere, therefore, it is unlikely that changes in the relative humidity will be the cause of large disparities between equilibrium partitioning theory and dynamic condensation.

The proposed method of solution is found to be exceptionally accurate for a wide range of parameters. Assuming the same mole fraction for all of the modes offers a quick method of obtaining a reasonably accurate approximation for all but the smallest van't Hoff factors of the involatile compounds but performs well at high values of relative humidity relevant to atmospheric applications. The perturbation correction term offers significant improvements at lower relative humidity, especially for smaller van't Hoff factors, with negligible increase in computational expense.

## 9 Code Availability

All Matlab source code used in the generation of the plots in this paper, including a solver for the multiple mode equilibrium partitioning theory equations, is available at dx.doi.org/10.5281/zenodo.34025.

## Appendix A

The implicit equations governing the leading order solution can be manipulated to give an explicit expression for each of the condensed concentrations, $\bar{C}_{ij}$, in terms of the average mole fraction. We present the algebra for such a step here. The coupled equations are given by (28) and are restated here

$$\bar{C}_{ij}^c = \frac{C_j - \sum_r \bar{C}_{rj}^c + \bar{C}_{ij}^c}{1 + \dfrac{C_j^*/\eta_i}{C_i^o(1+\beta)}}.$$

We can make use of the notation given by (14) to write the summation term as $\bar{C}_j^c$ and rearrange the denominator on the right-hand side to give

$$\bar{C}_{ij}^c = \left(C_j - \bar{C}_j^c + \bar{C}_{ij}^c\right) \frac{C_i^o(1+\beta)}{C_i^o(1+\beta) + C_j^*/\eta_i}.$$

The explicit dependence on $\bar{C}_{ij}^c$ can be factorised onto the left-hand side

$$\left(\frac{C_j^*/\eta_i}{C_i^o(1+\beta) + C_j^*/\eta_i}\right)\bar{C}_{ij}^c = \frac{C_i^o(1+\beta)\left(C_j - \bar{C}_j^c\right)}{C_i^o(1+\beta) + C_j^*/\eta_i},$$

which further reduces to

$$\bar{C}_{ij}^c = \frac{C_i^o(1+\beta)\left(C_j - \bar{C}_j^c\right)}{C_j^*/\eta_i}. \tag{A1}$$

By summing over $i$ an equation for $C_j^c$ is obtained

$$\bar{C}_j^c = \left(C_j - \bar{C}_j^c\right)\sum_r \frac{C_r^o(1+\beta)}{C_j^*/\eta_r},$$

which has the solution

$$\bar{C}_j^c = \phi_j C_j,$$

where $\phi_j$ depends on $\beta$ and $\eta_i$ and is given by

$$\phi_j(\beta,\eta_i) = \frac{\displaystyle\sum_r \frac{C_r^o(1+\beta)}{C_j^*/\eta_r}}{1 + \displaystyle\sum_r \frac{C_r^o(1+\beta)}{C_j^*/\eta_r}}, \tag{A2}$$

The individual condensed concentrations are then be calculated using (A1) which can now be writen as

$$\bar{C}_{ij}^c = \frac{C_i^o(1+\beta)(1-\phi_j)C_j}{C_j^*/\eta_i}.$$

## Appendix B

The details of the linearisation of the perturbation equations (30) are presented in this appendix and we begin by restating the set of non-linear equations

$$\bar{C}_{ij}^c + \hat{C}_{ij}^c = \frac{C_j - \sum_r \bar{C}_{rj}^c + \bar{C}_{ij}^c - \sum_r \hat{C}_{rj}^c + \hat{C}_{ij}^c}{1 + \dfrac{C_j^* \bar{K}_{ij}/\bar{\eta}_i}{C_i^o + \sum_k \bar{C}_{ik}^c + \sum_k \hat{C}_{ik}^c}}.$$

The denominator on the right-hand side can be rearranged to give

$$\bar{C}_{ij}^c + \hat{C}_{ij}^c = \left( C_j - \sum_r \bar{C}_{rj}^c + \bar{C}_{ij}^c - \sum_r \hat{C}_{rj}^c + \hat{C}_{ij}^c \right) \left( \frac{C_i^o + \sum_k \bar{C}_{ik}^c + \sum_k \hat{C}_{ik}^c}{C_i^o + \sum_k \bar{C}_{ik}^c + \sum_k \hat{C}_{ik}^c + C_j^* \bar{K}_{ij}/\bar{\eta}_i} \right)$$

$$= \left( \frac{C_j - \sum_r \bar{C}_{rj}^c + \bar{C}_{ij}^c - \sum_r \hat{C}_{rj}^c + \hat{C}_{ij}^c}{C_i^o + \sum_k \bar{C}_{ik}^c + C_j^* \bar{K}_{ij}/\bar{\eta}_i} \right) \left( \frac{C_i^o + \sum_k \bar{C}_{ik}^c + \sum_k \hat{C}_{ik}^c}{1 + \dfrac{\sum_k \hat{C}_{ik}^c}{C_i^o + \sum_k \bar{C}_{ik}^c + C_j^* \bar{K}_{ij}/\bar{\eta}_i}} \right)$$

$$= \left( \frac{C_j - \sum_r \bar{C}_{rj}^c + \bar{C}_{ij}^c - \sum_r \hat{C}_{rj}^c + \hat{C}_{ij}^c}{C_i^o + \sum_k \bar{C}_{ik}^c + C_j^* \bar{K}_{ij}/\bar{\eta}_i} \right) \left( C_i^o + \sum_k \bar{C}_{ik}^c + \sum_k \hat{C}_{ik}^c \right)$$

$$\times \left( 1 + \frac{\sum_k \hat{C}_{ik}^c}{C_i^o + \sum_k \bar{C}_{ik}^c + C_j^* \bar{K}_{ij}/\bar{\eta}_i} \right)^{-1}.$$

We now assume that the perturbations are sufficiently small that

$$\left| \frac{\sum_k \hat{C}_{ik}^c}{C_i^o + \sum_k \bar{C}_{ik}^c + C_j^* \bar{K}_{ij}/\bar{\eta}_i} \right| \ll 1, \tag{B1}$$

so that the third term on the right-hand side can be approximated by its Taylor series expansion of the form

$$\frac{1}{1+x} \approx 1 - x + O(x^2),$$

with $x$ equal to the term (B1),

$$\bar{C}_{ij}^c + \hat{C}_{ij}^c = \left( \frac{C_j - \sum_r \bar{C}_{rj}^c + \bar{C}_{ij}^c - \sum_r \hat{C}_{rj}^c + \hat{C}_{ij}^c}{C_i^o + \sum_k \bar{C}_{ik}^c + C_j^* \bar{K}_{ij}/\bar{\eta}_i} \right) \left( C_i^o + \sum_k \bar{C}_{ik}^c + \sum_k \hat{C}_{ik}^c \right)$$

$$\times \left( 1 - \frac{\sum_k \hat{C}_{ik}^c}{C_i^o + \sum_k \bar{C}_{ik}^c + C_j^* \bar{K}_{ij}/\bar{\eta}_i} + O\left( \left[ \hat{C}_{ij}^c \right]^2 \right) \right).$$

The right-hand side can be linearised assuming the terms of order $O\left( \left[ \hat{C}_{ij}^c \right]^2 \right)$ are negligible.

$$\bar{C}_{ij}^c + \hat{C}_{ij}^c = \frac{C_j - \sum_r \bar{C}_{rj}^c + \bar{C}_{ij}^c}{1 + \dfrac{C_j^* \bar{K}_{ij}/\bar{\eta}_i}{C_i^o + \sum_k \bar{C}_{ik}^c}} + \frac{-\sum_r \hat{C}_{rj}^c + \hat{C}_{ij}^c}{1 + \dfrac{C_j^* \bar{K}_{ij}/\bar{\eta}_i}{C_i^o + \sum_k \bar{C}_{ik}^c}}$$

$$+ \left( \frac{C_j - \sum_r \bar{C}_{rj}^c + \bar{C}_{ij}^c}{C_i^o + \sum_k \bar{C}_{ik}^c + C_j^* \bar{K}_{ij}/\bar{\eta}_i} \right) \sum_k \hat{C}_{ik}^c$$

$$- \frac{\left( C_j - \sum_r \bar{C}_{rj}^c + \bar{C}_{ij}^c \right) \left( C_i^o + \sum_k \bar{C}_{ik}^c \right)}{\left( C_i^o + \sum_k \bar{C}_{ik}^c + \dfrac{C_j^* \bar{K}_{ij}}{\bar{\eta}_i} \right)^2} \sum_k \hat{C}_{ik}^c.$$

We can factorise this to give

$$\hat{C}_{ij}^c + \frac{\sum_r \hat{C}_{rj}^c - \hat{C}_{ij}^c}{1 + \dfrac{C_j^* \bar{K}_{ij}/\bar{\eta}_i}{C_i^o + \sum_k \bar{C}_{ik}^c}} - \left[ \frac{\left( C_j - \sum_r \bar{C}_{rj}^c + \bar{C}_{ij}^c \right) \left( C_i^o + \sum_k \bar{C}_{ik}^c \right) \dfrac{C_j^* \bar{K}_{ij}}{\bar{\eta}_i}}{\left( C_i^o + \sum_k \bar{C}_{ik}^c + \dfrac{C_j^* \bar{K}_{ij}}{\bar{\eta}_i} \right)^2} \right] \sum_k \hat{C}_{ik}^c$$

$$= \frac{C_j - \sum_r \bar{C}_{rj}^c + \bar{C}_{ij}^c}{1 + \dfrac{C_j^* \bar{K}_{ij}/\bar{\eta}_i}{C_i^o + \sum_k \bar{C}_{ik}^c}} - \bar{C}_{ij}^c.$$

By denoting the coefficient in the square brackets by $\mathcal{L}_{ij}$ together with

$$\bar{\xi}_{ij} = \left( 1 + \frac{C_j^* \bar{K}_{ij}/\bar{\eta}_i}{C_i^o + \sum_k \bar{C}_{ik}^c} \right)^{-1}, \tag{B2}$$

this expresion can be made more notationally simplistic

$$680 \quad \left(1 - \bar{\xi}_{ij}\right) \hat{C}^c_{ij} + \bar{\xi}_{ij} \sum_r \hat{C}^c_{rk} - \mathcal{L}_{ij} \sum_k \hat{C}^c_{ij} = \frac{C_j - \sum_r \bar{C}^c_{rj} + \bar{C}^c_{ij}}{1 + \dfrac{C^*_j \bar{K}_{ij}/\bar{\eta}_i}{C^o_i + \sum_k \bar{C}^c_{ik}}} - \bar{C}^c_{ij}.$$

It is important to note that both $\bar{\xi}_{ij}$ and $\mathcal{L}_{ij}$ depend on the leading order solution but are independent of the perturbation, $\hat{C}_{ij}$, and as such this equation is now linear in these quantities.

*Acknowledgements.* The research leading to these results has received funding from the European Union's Seventh Framework Programme (FP7/2007-2013) under grant agreement n° 603445.

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

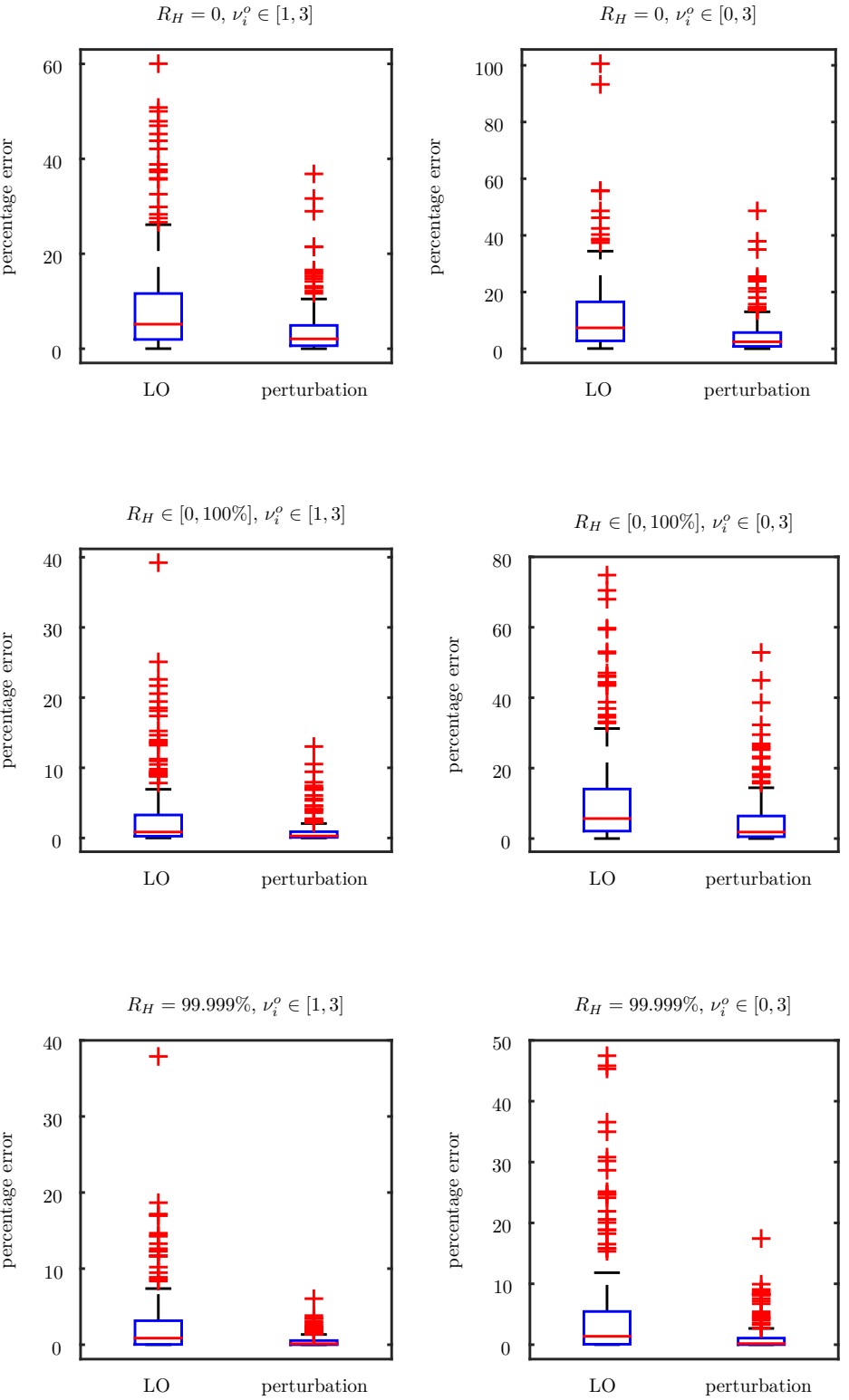

**Figure 8.** Comparison of the percentage errors in organic mass fraction from the two approximations in relation to those calculated using the non-linear solver for the data points in Figure 7. The leading order (LO) and perturbation solutions are shown by the left and right box and whisker plots in each figure, respectively.

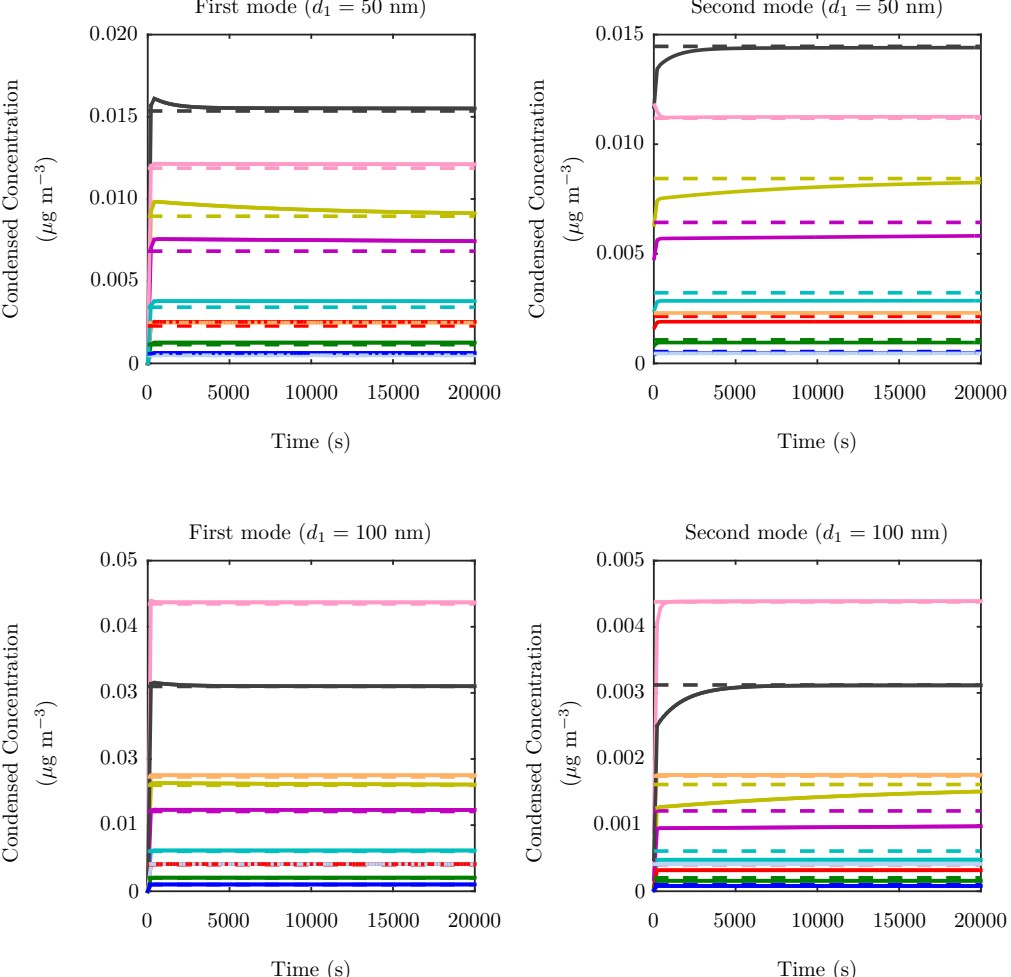

**Figure 9.** Time evolution of the total condensed concentrations of organics in each volatility bin in two runs of parcel model (solid lines). The top two plots shown the individual condensed concentrations on each of the modes when the smaller mode has a diameter of 50 nm and the lower two plots show the results when the smaller mode has a diameter of 100 nm. For both runs the number concentrations for the first mode was 1000 pcc and second mode was 50 pcc. Each colour represents a different volatility bin and the solution from the partitioning theory is shown by the dashed lines.

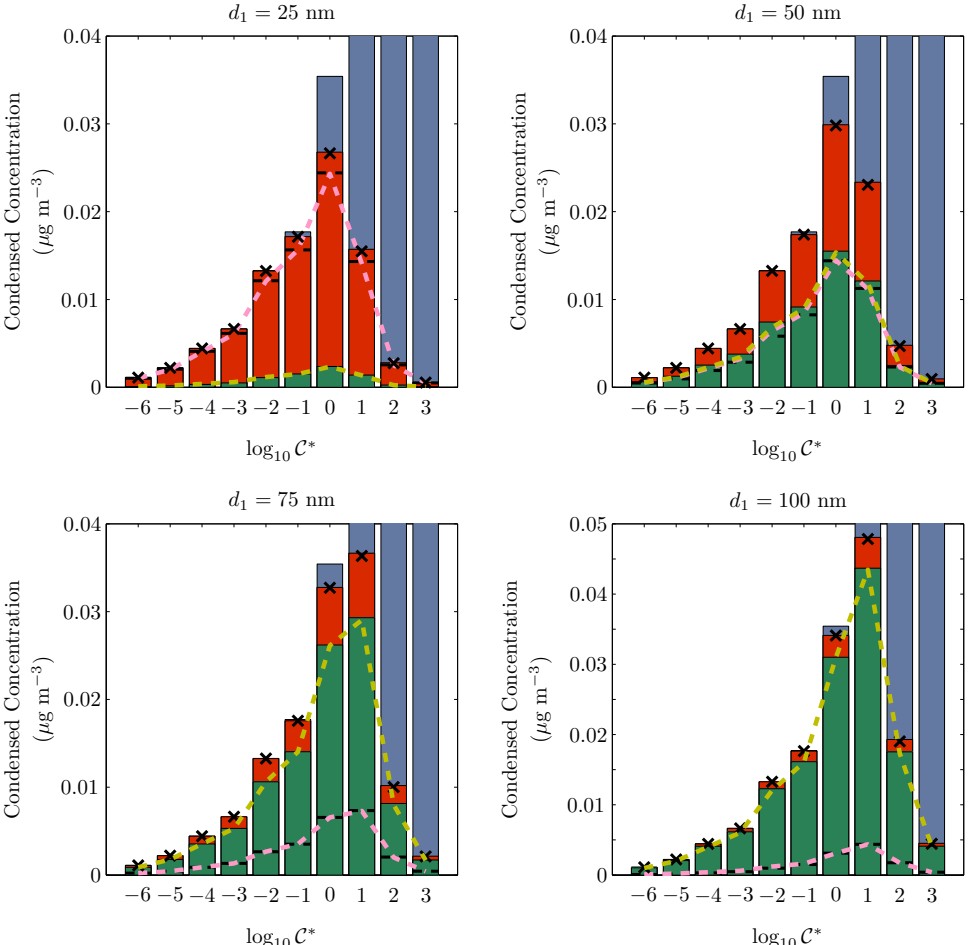

**Figure 10.** Stacked bar charts showing the condensed concentrations from the two models. Each bar shows the total concentration of organics in each volatility bin and are coloured to show the proportion which is in the vapour phase (blue) and the condensed phases on the first and second modes (green and red respectively). The height of the horizontal black dashed lines across each bar marks the condensed concentrations on the second mode only (the height of the red regions as measured from the $x$ axis). Total concentrations in each volatility bin from the partitioning theory are shown by the black crosses and the amount on the first and second modes are shown by the yellow and pink dashed lines respectively. The $y$ axis is cut off at 0.04 and 0.05 for clarity. The diameter of the first mode is denoted above each plot and the number concentration is 1000 pcc.

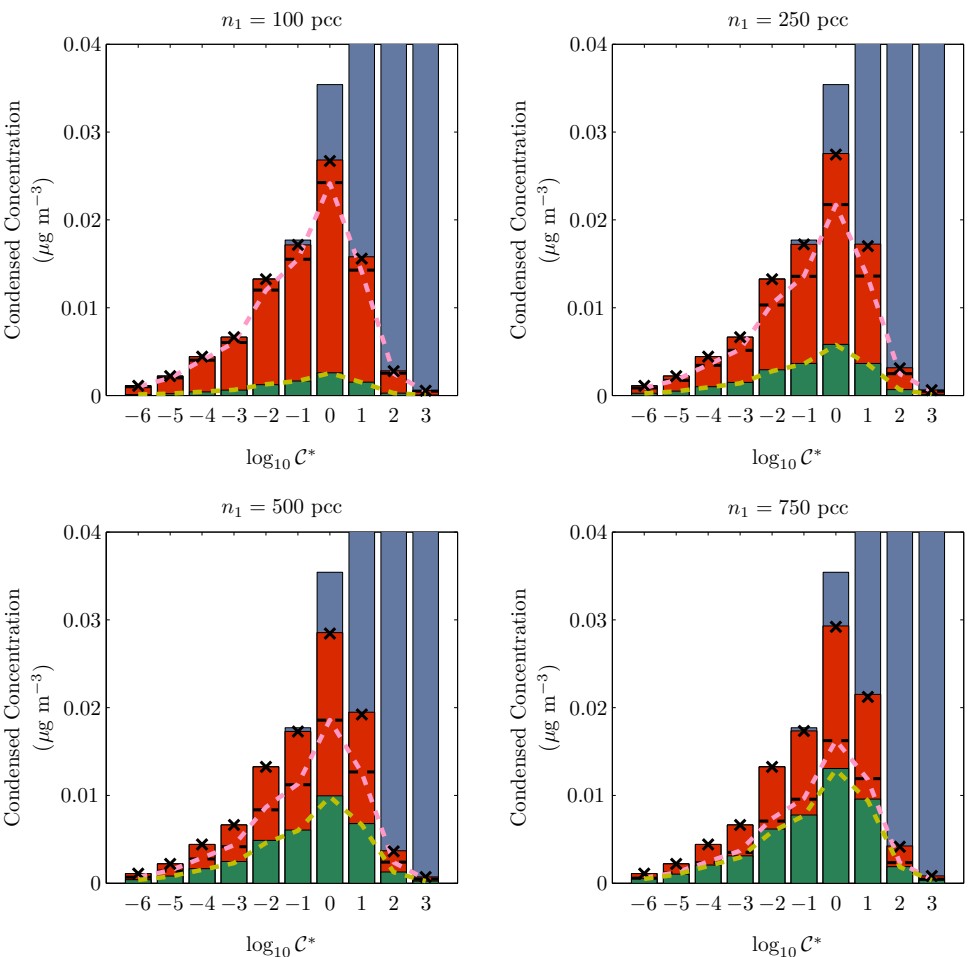

**Figure 11.** Same as Figure 10 but with differing number concentration in each plot. The diameter of the first mode is 50 nm.

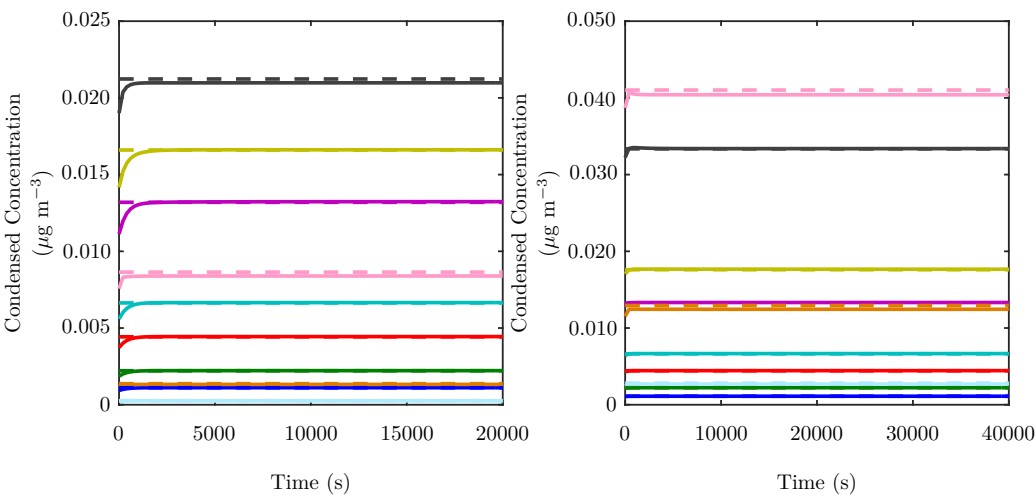

**Figure 12.** Time evolution of the total condensed concentrations of each organic in the parcel model (solid lines) for $R_H = 0\%$ (left) and $R_H = 90\%$ (right). Each colour represents a different volatility bin and the solution from the partitioning theory is shown by the dashed lines.

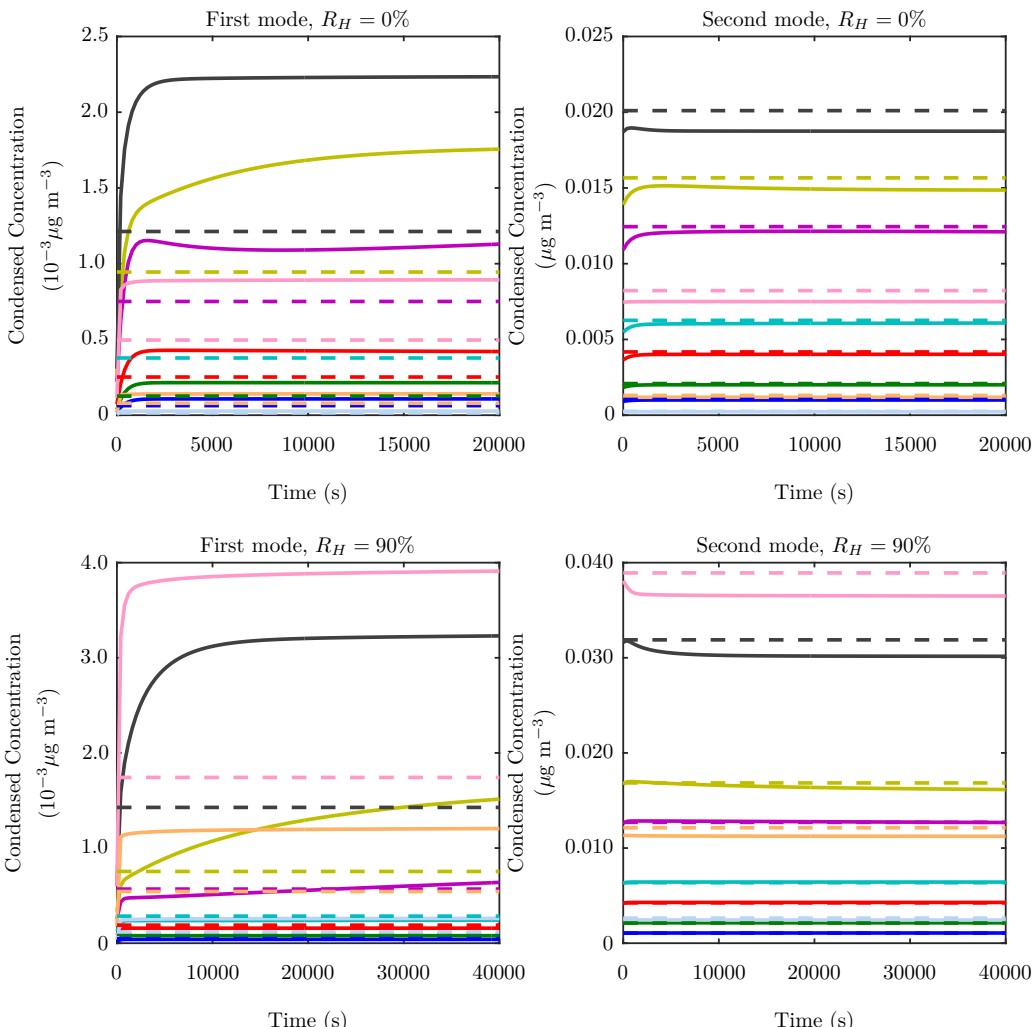

**Figure 13.** Time evolution of the condensed concentrations on each of the modes in the parcel model (solid lines) for $R_H = 0\%$ (top) and $R_H = 90\%$ (bottom). The smaller mode is shown on the left and the larger on the right. Each colour represents a different volatility bin and the solution from the partitioning theory is shown by the dashed lines.

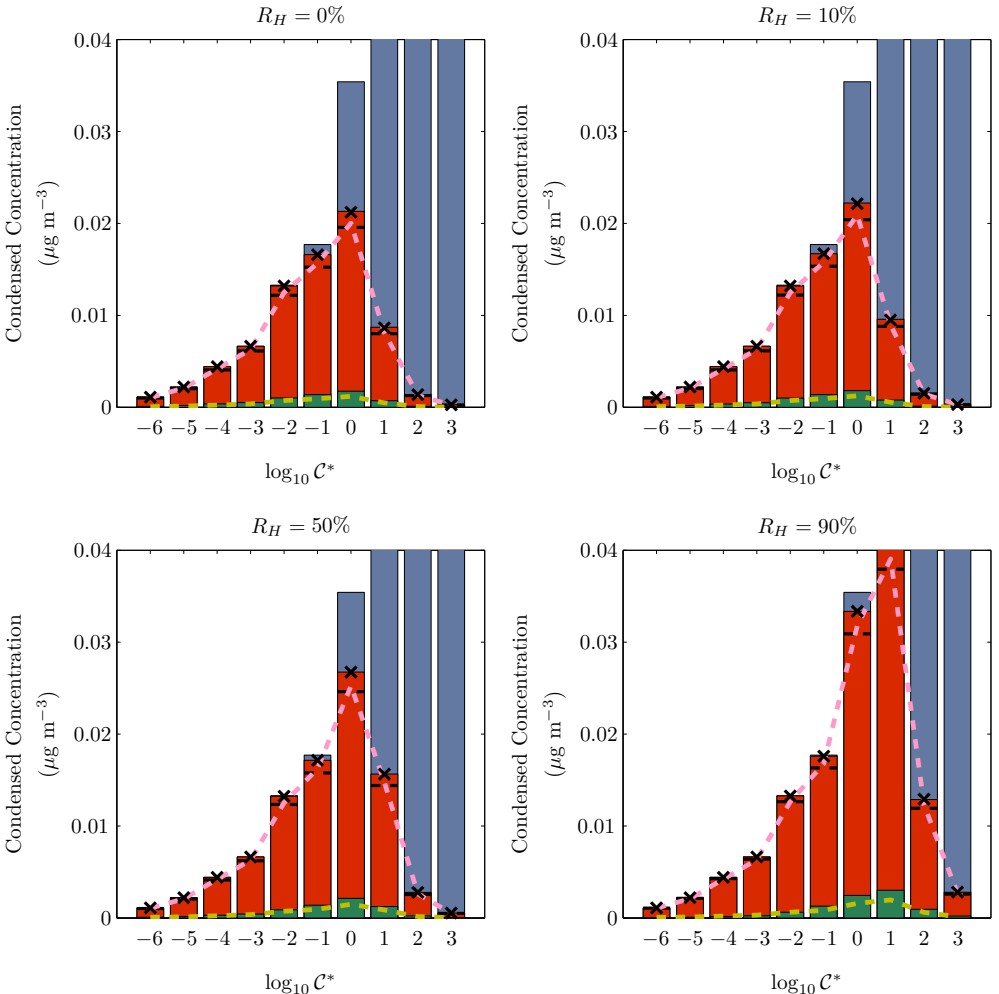

**Figure 14.** Same as Figure 10 for involatile aerosol modes represented by lognormal size distributions with number concentrations and median diameters given in Table 5. The relative humidity used in each simulation is stated above each plot.

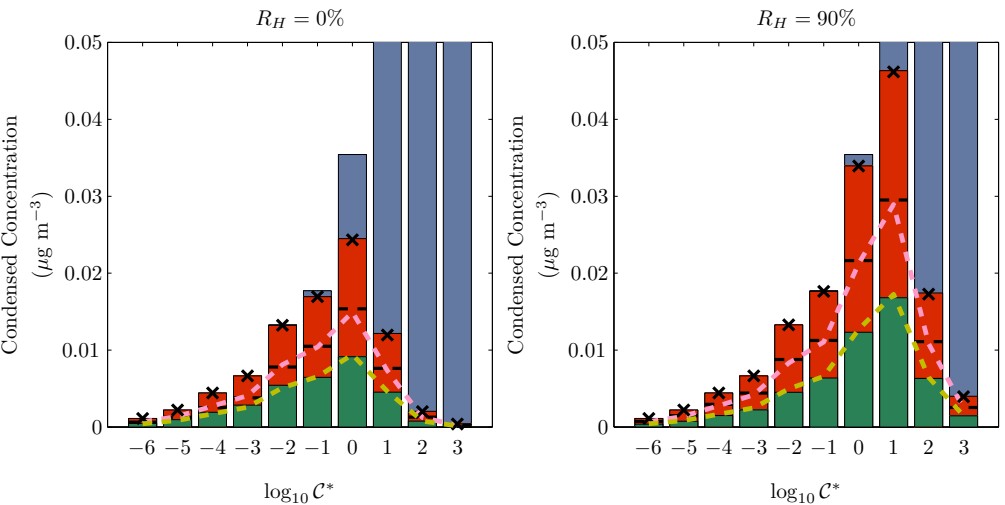

**Figure 15.** Same as Figure 14 but the first mode has a median diameter of 50 nm.

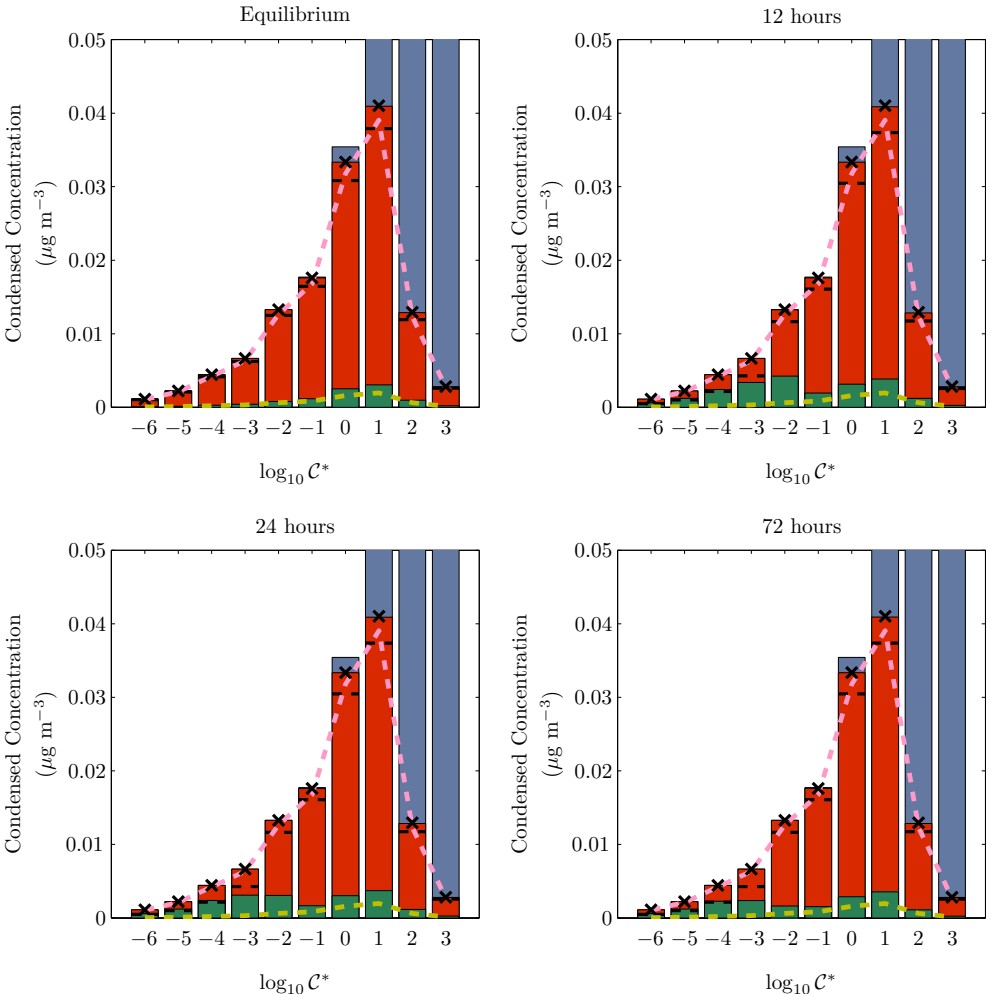

**Figure 16.** Comparison between condensed mass calculated using equilibrium partitioning theory against the dynamic condensation solution at a range of times. Aerosol parameters are the same as in Figure 14 and the relative humidity is 90%.

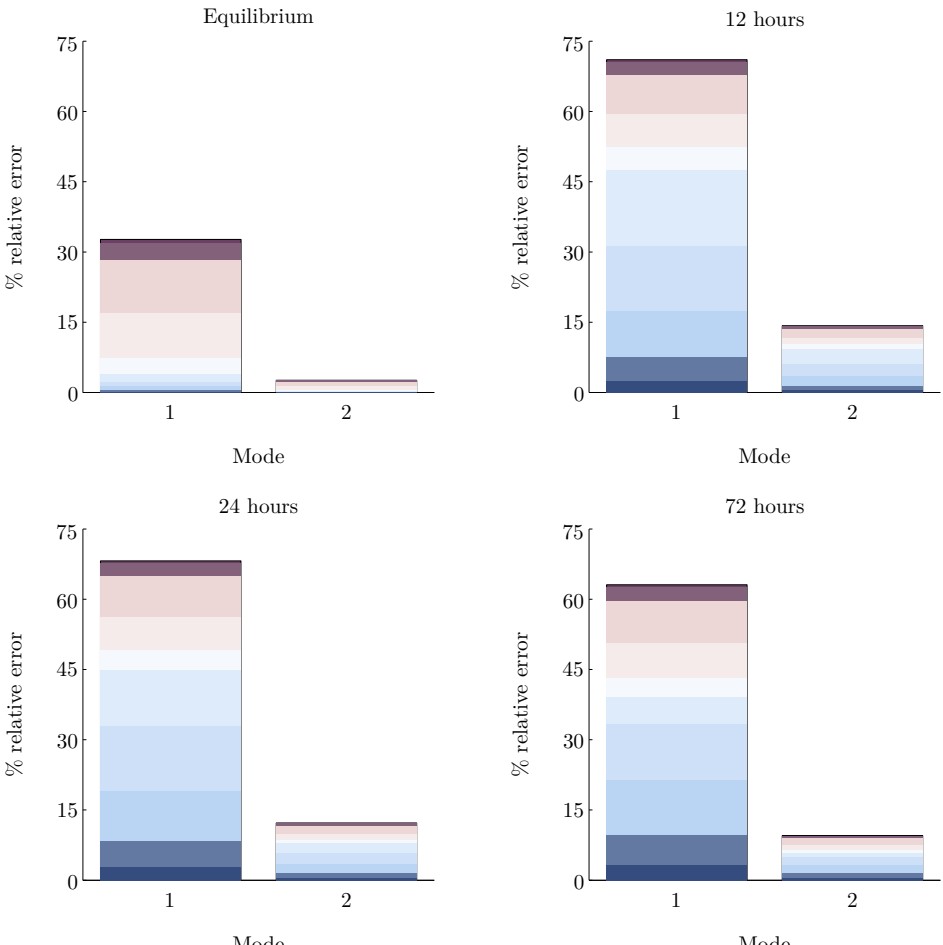

**Figure 17.** Relative error of the condensed mass from equilibrium partitioning theory onto 2 lognormal size distributions compared against the dynamic solution. Each subfigure corresponds to plots in Figure 16. The bars are subdivided to show the contribution from each volatility bin and are coloured with low $C^*$ values in dark blue and high $C^*$ values in dark purple.

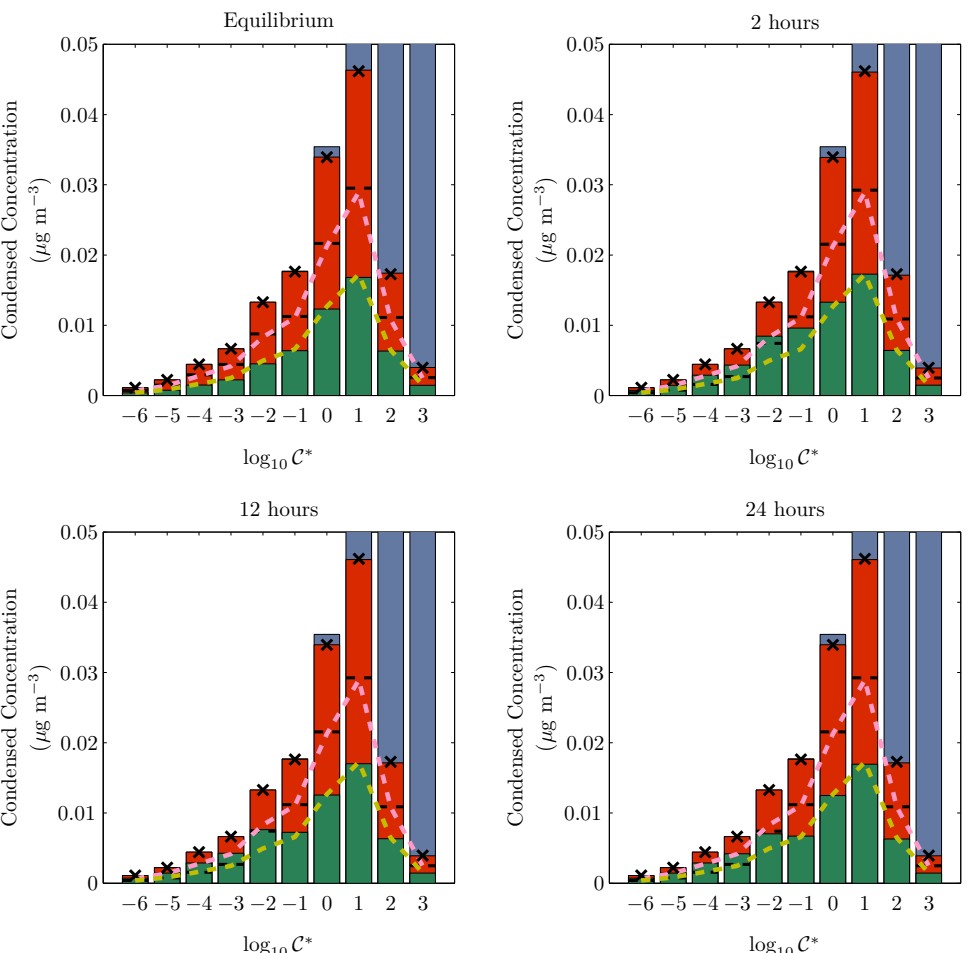

**Figure 18.** Same as Figure 16 but the first mode has a median diameter of 50 nm.

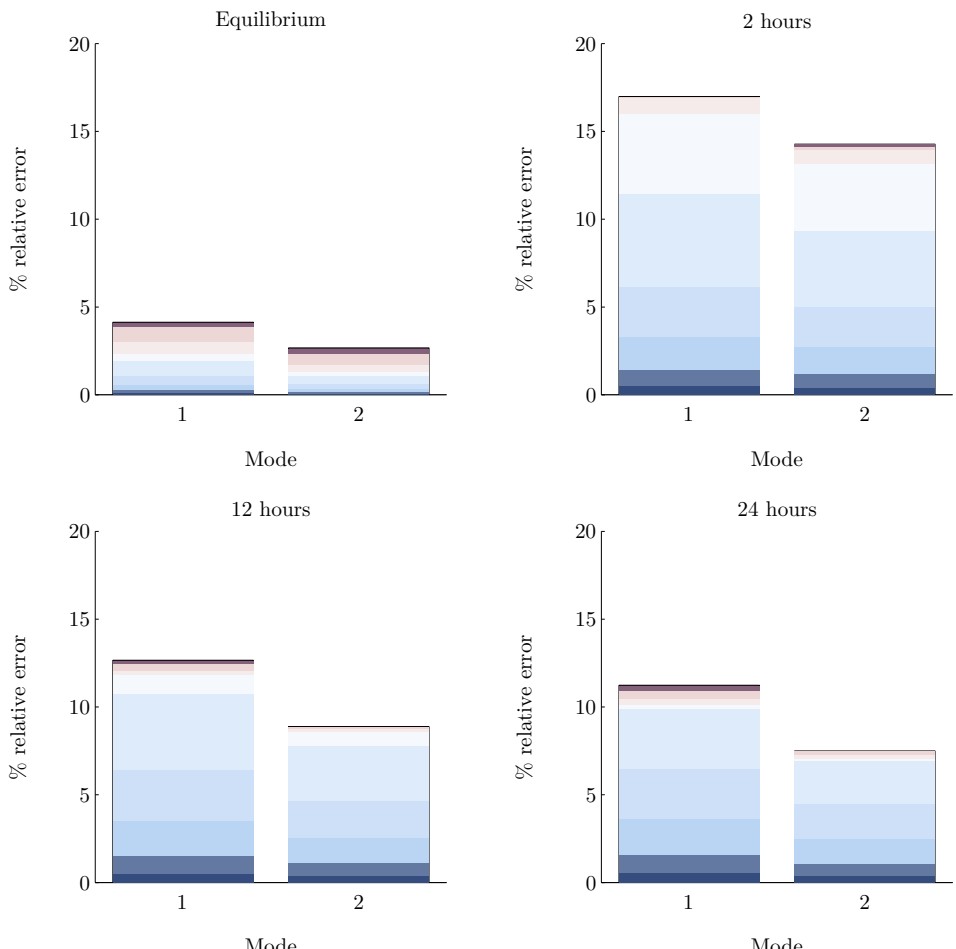

**Figure 19.** Same as Figure 17 but showing the errors in Figure 18.

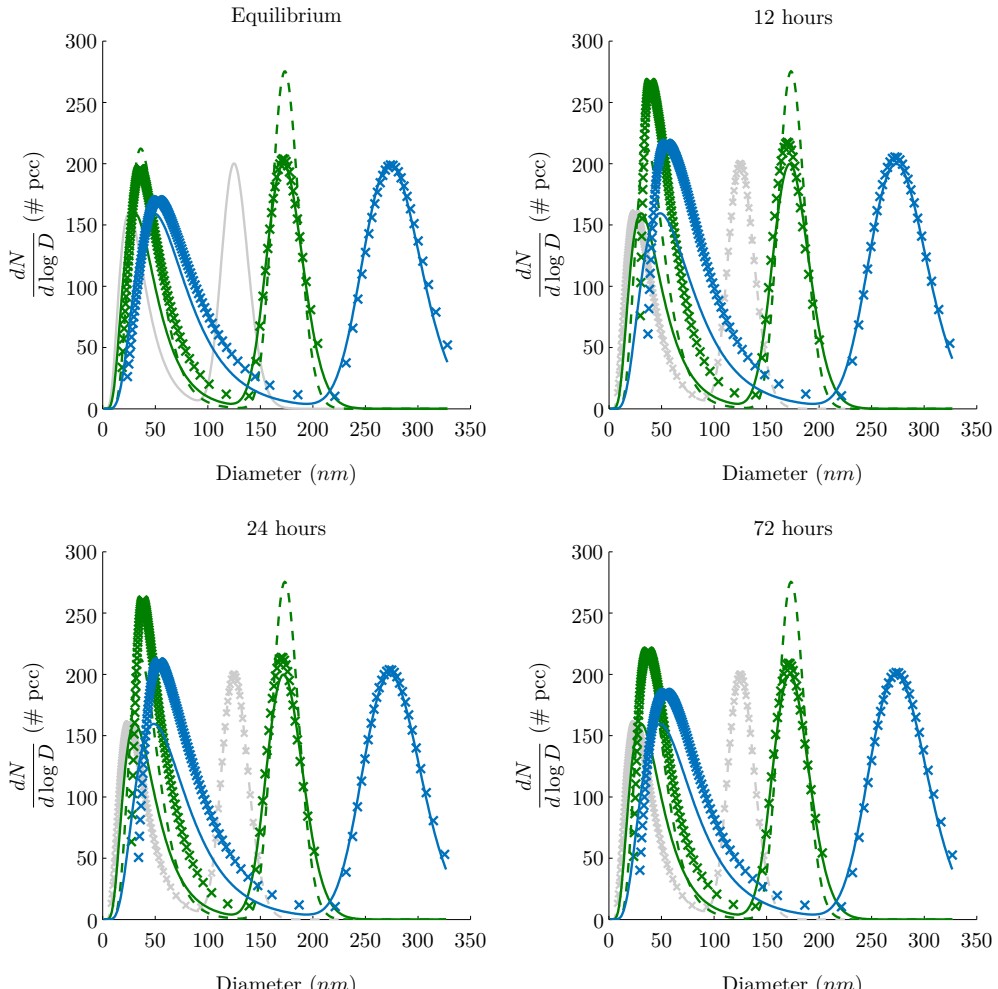

**Figure 20.** Time evolution of the size distribution. The grey line shows the initial size distribution and the blue and green show the wet and dry aerosol size distributions, repectively. The crosses are from the parcel model, the solid lines are from equilibrium partitioning assuming constant geometric standard deviation and the dashed lines are calculated maintaining a constant arithmetic standard deviation.

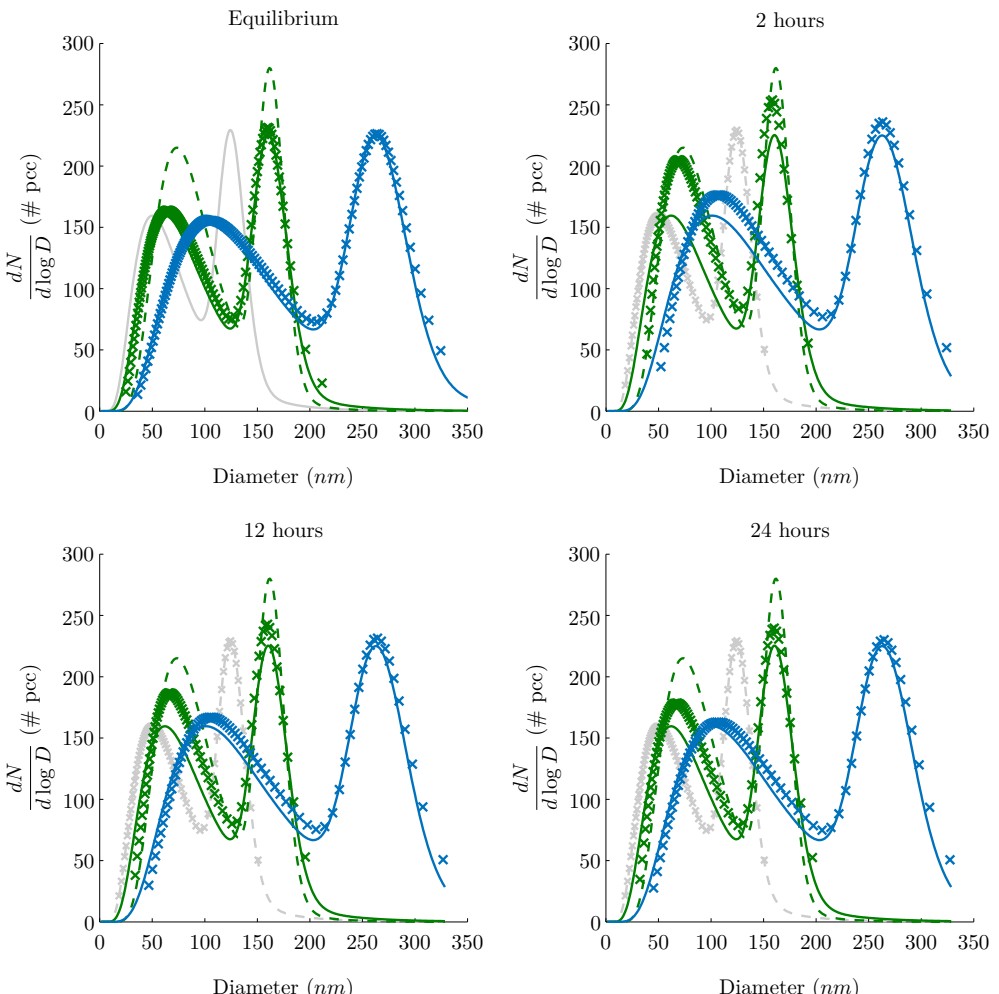

**Figure 21.** Same as Figure 20 with the first mode of median diameter 50 nm.