# Peer review of "The co-condensation of semi-volatile organics into multiple aerosol particle modes"

_Geoscientific Model Development, 2015_

## Referee Comment (RC1) · Anonymous Referee #1 · 4 Mar 2016

Solving the partitioning of semivolatile compounds between the gas phase and different sized particles is a challenging task. In this manuscript authors provide a new method for solving the equilibrium partitioning analytically. The method is thoroughly presented and compared against numerical solution. Overall, the topic of the manuscript is suitable for Geoscientific Model Development and the method presented might be valuable for scientific community using volatility basis set to represent organic aerosol in large scale models. I recommend the manuscript to be accepted after the comments below have been addressed.

My main criticism is related to examples and discussion presented in Chapter 7. After reading the manuscript the main question is if the parameterization is good enough in representing dynamic conditions so that it can be used in atmospheric applications? It is difficult to understand what is done in Chapter 7. What the modelling setup described

at lines 390-394 actually mean? Does it mean that for the first 300 or 10 seconds the condensation is only allowed for the second mode and after that also for the first mode? Maybe more discussion motivating the modelling setup is needed to make it easier to understand. One way to estimate if equilibrium approximation gives reasonable results in atmospheric conditions would be to run the parcel model with varying temperature and relative humidity trajectories mimicking the turnover of boundary layer.

Minor comments:

I think it is quite uncommon to use $R\_H$ notation for saturation ratio. I would recommend changing it to S in equations as it is commonly used and because you are also discussing RH as relative humidity, which is usually expressed in percentage.

Line 49: Citation to Connolly et al. (2014) might be in a wrong place. At least I cannot find relevant discussion on aerosol forcing from that article.

Line 70: Why is Hildemann et al. (1991) article cited when discussing on the current status of knowledge on atmospheric organic components. Has there been any progress since 1991?

At least for me, the "Quasi-leading order solution" is a totally new term for describing the solution. Why "quasi"?

Figure 3: How many solutions there is actually presented?

Figure 9: Diameters are really confusing as they are different than given in the Table 4 and in the text.

Figure 13: It looks like the system is not evolving towards equilibrium conditions. Why not?

Figure 15: The change in the shape of aerosol size distribution in numerical solution is strange. Is there some reason why larger particles grow more so that the shape of distribution is not preserved?

[Figure]

---

## Referee Comment (RC2) · Anonymous Referee #2 · 22 Mar 2016

Review of Crooks et al. "The co-condensation of semi-volatile organics into multiple aerosol particle modes. The authors have presented a new methodology and detailed set of non-linear equations to solve the co-condensation of organics into multiple modes. Although these mathematical equations and their derivations are presented in great details, their utility and atmosphere relevance is not clear. My major comment is that even after several reads, the paper mostly sounds like new and fairly involved algebraic mathematical formulations which are interesting, but why should atmospheric scientists care about these formulations? Below are specific comments that need to be addressed before the manuscript is considered for publication. 1. The condensation of semi-volatile organics on multiple modes is not a new formulation. This has been done in other models [e.g., Liu et al., 2012]. In the previous formulation [e.g., Liu et al., 2012], the sum of a semi-volatile organic partitioned to various modes equals the total

aerosol particle fraction as determined by gas-particle partitioning theory. Therefore, it was not immediately clear how the authors work differs from multi-mode partitioning of semi-volatile organics in previous studies, except that it includes water and a non-volatile core. The authors need to clearly make this distinction between their work and previous multi-mode partitioning studies. 2. The title mentions co-condensation of semi-volatile organics. But it's actually co-condensation of organics and water on non-volatile core aerosol. The title needs to better reflect what is being presented. 3. What is the composition of the non-volatile core? Does it include inorganics such as sulfate, nitrate and also black carbon and non-volatile organic aerosol? The authors need to clearly define the composition of the core aerosol. 4. If the core aerosol includes inorganics, the authors are implicitly assuming that the inorganic core aids the partitioning of semi-volatile organics e.g. see equation 3, where the non-volatile core is included in the calculation of COA. How is this assumption justified? The absorptive partitioning theory assumes well mixed solution [Pankow, 1994]. How can a core-shell model be well mixed? Also, several studies suggest that secondary organic aerosols (SOA) are under many conditions highly viscous semi-solids [Cappa and Wilson, 2011; Vaden et al., 2011; Virtanen et al., 2010], so they cannot be assumed to be well mixed. The authors need to clearly specify where their current formulation is not atmospherically relevant in the context of these studies. 5. Section 7, page 18: The authors place large particles in the model first before adding small particles to improve the accuracy of their solution. How can this be applied in a regional or global 3D model, where many processes are happening simultaneously (such as nucleation, emissions, coagulation, condensation, transport etc.) so that at any time there are both small and big particles? 6. Finally, does the author's new formulation include organic-inorganic interactions especially for aqueous aerosols? For example, I did not see hygroscopicity of the core and other organics include anywhere in their equations. Please clarify how the differing hygroscopicities, activities and aqueous phase reactions would affect your equations and their solutions.

References: 1. Cappa, C. D., and K. R. Wilson (2011), Evolution of organic aerosol

mass spectra upon heating: implications for OA phase and partitioning behavior, Atmos. Chem. Phys., 11(5), 1895-1911. 2. Liu, X., et al. (2012), Toward a minimal representation of aerosols in climate models: description and evaluation in the Community Atmosphere Model CAM5, Geosci. Model Dev., 5(3), 709-739. 3. Pankow, J. F. (1994), An absorption model of the gas aerosol partitioning involved in the formation of secondary organic aerosol Atmospheric Environment, 28(2), 189-193. 4. Vaden, T. D., D. Imre, J. Beranek, M. Shrivastava, and A. Zelenyuk (2011), Evaporation kinetics and phase of laboratory and ambient secondary organic aerosol, Proc. Natl. Acad. Sci. U. S. A., 2190-2195. 5. Virtanen, A., et al. (2010), An amorphous solid state of biogenic secondary organic aerosol particles, Nature, 467(7317), 824-827.

---

## Author Comment (AC1) · 29 Apr 2016

1. "My main criticism is related to examples and discussion presented in Chapter 7. After reading the manuscript the main question is if the parameterization is good enough in representing dynamic conditions so that it can be used in atmospheric applications? It is difficult to understand what is done in Chapter 7. What the modelling setup described at lines 390-394 actually mean? Does it mean that for the first 300 or 10 seconds the condensation is only allowed for the second mode and after that also for the first mode? Maybe more discussion motivating the modelling setup is needed to make it easier to understand. One way to estimate if equilibrium approximation gives reasonable results in atmospheric conditions would be to run the parcel model with varying temperature and relative humidity trajectories mimicking the turnover of boundary layer."

Multiple mode equilibrium partitioning is designed for multiple monodisperse modes. The aim of the Chapter 7 was to investigate how well the theory works if you use log-normal size distributions with effective Kelvin terms (using the median diameter). The reason we allowed condensation to occur only on the larger particles initially was to speed up the equilibration process to allow us to compare the how well equilibrium partitioning theory can deal with the lognormal size distributions rather than this mimicking the lifetime of aerosol particles and SVOCs in the atmosphere.

In order to address this issue we have added a specific section 7.2 to show the existing results and have tried to clarify the purpose of the section more clearly. Section 7.3 has additionally been added to compare how the equilibrium model compares to the parcel model when all particles and SVOCs are allowed to interact simultaneously from the being of the simulation.

2. "I think it is quite uncommon to use R_H notation for saturation ratio. I would recommend changing it to S in equations as it is commonly used and because you are also discussing RH as relative humidity, which is usually expressed in percentage."

"R_H" changed to "S" where appropriate.

3. "Line 49: Citation to Connolly et al. (2014) might be in a wrong place. At least I cannot find relevant discussion on aerosol forcing from that article."

Citation removed

4. "Line 70: Why is Hildemann et al. (1991) article cited when discussing on the current status of knowledge on atmospheric organic components. Has there been any progress since 1991?"

Citation updated

5. "At least for me, the "Quasi-leading order solution" is a totally new term for describing the solution. Why "quasi"?"

"Leading order" has a precise mathematical definition which is slightly different from what we have used, hence the "quasi" in front. However, given the readership of GMD it is probably over pedantic to add the "quasi" so we have removed it.

6. "Figure 3: How many solutions there is actually presented?"

50: previously it stated 30 but there are actually 50.

7. "Figure 9: Diameters are really confusing as they are different than given in the Table 4 and in the text."

We think they are the same. Table 4 shows the range of values used in Figures 9-11 and Figure 9 shows only some of these values. Is the confusion that the title contains the diameter of the first mode? ie "d1 = 50 nm". This is necessary to distinguish between the two sets of results, one with a small mode of diameter 50nm and the other with a small mode diameter of 100 nm. It was a little confusing in the right two plots for this to follow on from the words "second mode" so we have added brackets around the "d1 = 50 nm". We have also changed the upper case notation for diameter to a lower case to be consistent with the rest of the text. The number concentration is now explicitly written in the figure caption too.

8. "Figure 13: It looks like the system is not evolving towards equilibrium conditions. Why not?"

It is not evolving towards to equilibrium. This section was intended to compare how well equilibrium partitioning (designed for monodisperse modes) could cope with each mode being a lognormal size distribution. It turns out that there can be discrepancies for size distributions with median diameters of less than about 50nm. This is discussed further in the response to the first comment.

9. "Figure 15: The change in the shape of aerosol size distribution in numerical solution is strange. Is there some reason why larger particles grow more so that the shape of distribution is not preserved?"

We presume you mean the linear section to the right of the size distribution. This is a result of using a bin structure with the same number of particles in each bin. At large sizes you can end up with a very wide bin and that linear section is simply Matlab joining two data points by a straight line. The size distribution plots have been re-done with 140 size bins in response to the first comment, which has resolved this issue.

Please also note the supplement to this comment:
http://www.geosci-model-dev-discuss.net/gmd-2015-187/gmd-2015-187-AC1-supplement.pdf

**Supplement:**

[revised manuscript text omitted]

---

## Author Comment (AC2) · 29 Apr 2016

0. "The co-condensation of semi-volatile organics into multiple aerosol particle modes. The authors have presented a new methodology and de- tailed set of non-linear equations to solve the co-condensation of organics into multiple modes. Although these mathematical equations and their derivations are presented in great details, their utility and atmosphere relevance is not clear. My major comment is that even after several reads, the paper mostly sounds like new and fairly involved algebraic mathematical formulations which are interesting, but why should atmospheric scientists care about these formulations?"

The paper initially presents a mathematical formulation for equilibrium absorptive partitioning theory to be used as an approximation for the dynamically condensation process that occurs in the atmosphere. The advantage of such a formulation is the algebraic nature of the equations rather than the differential equations that result from dynamic condensation. These algebraic equations depend on the instantaneous ambient conditions and do not need to be solved sequentially in time. Thus making their solution much less computationally expensive and therefore, when used in GCMs, can significantly reduce the computational expense. Even in situations when it is possible to dedicate sufficient time and computing power to solve the time dependent dynamic solution it is not necessarily the best route to take; quicker methods can allow more simulations to be carried out and therefore offer a better insight into the problem. We also do not believe that it is possible to solve every physical process using the most "accurate" (or complicated) means and some process always has to be lost or simplified in order to focus in on one or a few of particular interest.

Due to the non-linear nature of the equations, obtaining a solution is still a difficult task to carry out numerically. Non-linear solvers exist but require initial guesses for the solution and the more dimensions or unknowns that exist in the problem the better this guess has to be. In the current problem initial guess of zeros or ones is far from sufficiently accurate for any solver to converge and in a best case scenario will only find a local minimum. Starting with an initial guess far from the correct solution also increases the computational expense of the solution. The leading order solution offers a way to reduce the number of unknowns to 1. Single dimensional solvers are far more robust and require much less accurate initial conditions. The solution can additionally be calculated in a fraction of the time it takes to run a multidimensional solver. This solution, once found, can then be used as an initial guess for the multidimensional solver, therefore reducing computational expense. We additionally explore, however, how accurate this initial guess is as an approximation to the solution in its own right and find that in a range of scenarios the errors compared to the solution to the multidimensional solver solution are as low as 10-15% (see figure 8). This may be sufficiently accurate for many applications.

We further suggest an alternative method to improving the accuracy of the leading order solution by perturbing the variables. Thus offering a semi-analytic method of increasing the accuracy without needing to use a multidimensional solver at all. This solution offers much improved accuracy (see figure 7) with greatly reduced computational expense compared to the multidimensional solver.

In summary, the applicability to atmospheric scientists is that our equilibrium partitioning may be able to replace hugely computationally expensive differential equation solvers in a range of work. In the case of GCMs this could make problems that have previously been thought of as too computationally expensive obtainable. In others, it may allow many more simulations to be carried out to span a wider parameter space and model many different scenarios.

1. "The condensation of semi-volatile organics on multiple modes is not a new formulation. This has been done in other models [e.g., Liu et al., 2012]."

It is not clear how [Liu et al., 2012] use multiple mode equilibrium partitioning. In section S1.1.3. SOA of their supplementary material they state "The condensation/evaporation is treated dynamically, as described below." – This suggests that the partitioning occurs dynamically and is not referencing an equilibrium partitioning, as proposed in the current work. Further to this, in Section S1.1.5. Condensation it says "Condensation of . . . the semi-volatile organics to various modes is treated dynamically, using standard mass transfer expressions" – this is similar to the dynamic parcel model that we are using to verify that our equilibrium partitioning theory calculates the correct condensed concentrations. Again, this is not an equilibrium partitioning theory.

We have changed "co-condensation" in line 85 of the original manuscript to "equilibrium absorptive partitioning" to try to stress the equilibrium aspect of the current work.

"The authors need to clearly make this distinction between their work and previous multi-mode partitioning studies"

We still maintain that this is the first representation of multiple mode equilibrium absorptive partitioning and a review of existing equilibrium partitioning is presented. To further draw attention to the equilibrium aspect of this work and, in particular, the crucial element of the involatile constituent in the particles which makes this possible we have added the penultimate paragraph in the introduction.

2. "The title mentions co-condensation of semi-volatile organics. But it's actually co-condensation of organics and water on non-volatile core aerosol. The title needs to better reflect what is being presented. "

We don't understand this comment; it's not clear whether the issue is with the use of "co-condensation", "semi-volatile", multiple organics/water or the non-volatile component in the aerosol particles.

3. "What is the composition of the non-volatile core? Does it include inorganics such as sulfate, nitrate and also black carbon and non-volatile organic aerosol? The authors need to clearly define the composition of the core aerosol."

The core aerosol can take any composition but we assume that it is soluble through the van't Hoff factor; low solubility compounds such as black carbon can be modelled by using values close to zero. We also assume that it contributes to the partitioning of the SVOCs and water. In our particular examples we use ammonium sulphate and this is reflected in the material properties stated in table 3.

4. "If the core aerosol includes inorganics, the authors are implicitly assuming that the inorganic core aids the partitioning of semi-volatile organics e.g. see equation 3, where the non-volatile core is included in the calculation of COA. How is this assumption justified? The absorptive partitioning theory assumes well mixed solution [Pankow, 1994]. How can a core-shell model be well mixed? Also, several studies suggest that secondary organic aerosols (SOA) are under many conditions highly viscous semi-solids [Cappa and Wilson, 2011; Vaden et al., 2011; Virtanen et al., 2010], so they cannot be assumed to be well mixed. The authors need to clearly specify where their

current formulation is not atmospheri- cally relevant in the context of these studies."

We do indeed assume that the core aerosol aids partitioning and we also assume the particles are well mixed. Perhaps the confusion lies in the use of "core aerosol"; we are referring to the non-volatile constituent that exists before condensation, rather than a shell/core model with diffusion. We have changed references to "core" to "involatile"

A discussion of particle viscosity has been added in the third to last paragraph of the introduction.

5. "Section 7, page 18: The authors place large particles in the model first before adding small particles to improve the accuracy of their solution. How can this be applied in a regional or global 3D model, where many processes are happening simultaneously (such as nucleation, emissions, coagulation, condensation, transport etc.) so that at any time there are both small and big particles?"

This has been answered in detail in response to the first referee's first comment.

6. "Finally, does the author's new formulation include organic-inorganic interactions es- pecially for aqueous aerosols? For example, I did not see hygroscopicity of the core and other organics include anywhere in their equations. Please clarify how the differing hygroscopicities, activities and aqueous phase reactions would affect your equations and their solutions."

The hygroscopicity, $\kappa$, as defined by [petters and kreidenweis 2007] can be used to derive equivalent values to the van't Hoff factor, $\upsilon$, and vice versa. As such the hygroscopicity can incorporated that way. We do not consider aqueous phase or gas phase reactions in our calculations.

Please also note the supplement to this comment:
http://www.geosci-model-dev-discuss.net/gmd-2015-187/gmd-2015-187-AC2-supplement.pdf

**Supplement:**

[revised manuscript text omitted]

---

## Editor Decision (ED1)

Topical editor review of Crooks et al. "The co-condensation of
semi-volatile organics into multiple aerosol modes".

The authors responses and associated revisions have addressed the issues
identified by the two reviewers, and the paper is now close to being
acceptable to proceed to publication.

However, despite the manuscript being much improved, my opinion as Topical
Editor, is that there are a few key issues that need clarifying and parts
of the manuscript are not well written, even in the revised version.
The paper is not yet of a high enough standard to warrant publication.

For that reason, I am recommending the authors make a further revised
version responding to the points I make in this Topical Editor review.

After reading the revised manuscript, it was clear that there are a
considerable number of minor revisions required and I list these below.

Provided these issues can be addressed, the paper could be suitable
for publication. However, I will need to see the paper again after the
authors have revised it, to consider whether or not it can be signed
off for publication in GMD.

In addition to the minor revisions, there are several major issues.

First, both reviewer 2, and my initial review identified an issue
with the term "co-condensation".

There seems to have been some confusion during the reviews, which I
believe to be mostly caused by the use of the term "co-condensation".

In one of their responses to reviewer 2 (the one numbered 1), the
authors explain they decided to change an occurrence of the term
"co-condensation" (on line 85) with the term "equilibrium absorptive
partitioning".

In my initial Topical Editor review I suggested the authors consider
replacing the term "co-condensation" with the more general term
"partitioning". My view there was that the term "co-condensation"
was not really appropriate terminology for semi-volatile species
(because it implies only one-way gas-to-particle transfer).
I suggested the term partitioning to avoid any potential confusion whereby
the reader might initially assume the method follows a kinetic approach.

I agree that using the longer and more descriptive term
"equilibrium absorptive partitioning" represents a further
improvement over the single-word term "partitioning".

However, although the authors have revised that particular instance
of the use of the term, the title of the paper still uses the term
"co-condensation".

For the reasons given above, I strongly recommend that the authors
avoid using that term "co-condensation".

In response to my querying the term in my initial editor review,
the authors did not defend its use, instead stating

"It wasn't clear whether "co-condensation" in the title was
ok or not."

When considering that response, I chose not to press the authors
to revise the term, instead leaving the term in to be considered

later during the main review process.

Since reviewer 2 has also requested that change, I am here
requesting the authors replace the term with "equilibrium
absorption partitioning" in all instances in the text, including
the title.

The 2nd major comment I have is regarding the two restrictions
that are placed on the method as it is being tested here.
See my two points 40 and 43 -- there needs to be stated clearly
some discussion of how these two restrictions limit the method's
applicability in a global model. How dependent is the method
on these two restrictions?

I need to be convinced this is properly discussed and considered
before allowing the paper to proceed to publication in GMD.

The 3rd point is regarding the section 7.3.2 which requires quite
some improvement (see my points 46 to 49)

The final point I make is about the conclusions which need to be
improved considerably with some discussion about the
limitations of the method and its applicability in a global model
(see my point 50).

It is for these latter 3 reasons that mean that I need to ask to
see the paper again after these revisions have been made before
I can decide whether or not it can proceed to publication.

Specific minor revisions

1) Abstract, Page 1 line 1 -- insert the word "absorptive" between
"equilibrium" and "partitioning" to match the terminology used
in response to reviewer 2's comment (see above).  Also replace
"gas/particle concentrations" with "gas and particle concentrations".
In this instance the partitioning is determining both the gas and
the particle concentrations, so both terms should be stated --
it is not a case of one or the other.

2) Abstract, Page 1, line 4 -- insert comma between "problem" and "the"

3) Abstract, Page 1, line 5 -- replace "on the aerosol" with "of the aerosol"

4) Abstract, Page 1, line 7 -- I found the term "pivotal" here not quite
appropriate -- to me the other assumption about the organic mole fraction
in the partitioning coefficient being across all modes seemed a more
pivotal assumption that enabled the equations to be approximated by a
simpler solution.  In fact I recommend to re-order the sentences with
the sentence beginning "The resulting coupled non-linear system" being
moved up to follow after the sentence ending "on each involatile mode".
The text seemed to flow well with that ordering, and it seemed to me
that the primary assumption was the organic mole fraction -- with the
part about each mode containing some involatile core (so never encountering
complete evaporation) being secondary.  I strongly suggest to replace the
words "The pivotal" with "Another key" and move that entire sentence to
come after the sentence ending "is set to be equal across all modes."
In that sentence referring to the involatile constituent I also
recommend to delete "to create a monodisperse aerosol at equilibrium"
and instead finish that sentence as "thereby avoiding associated
numerical difficulties" or similar. That "monodisperse aerosol"
didn't make sense to me -- but replacing to something like that
referring to the numerical issues avoided would work I think there.

5) Introduction, page 2, line 27 -- referencing style is incorrect
here (and in similar instances of several refs in brackets).
Need to revise to avoid double-brackets -- use /citep if
in LaTeX rather than /citet.

6) Introduction, page 2, line 40 -- Insert "than smaller ones" after
"water vapour more quickly".

7) Introduction, page 2, line 40 -- Replace "Consequently, this can
suppress..." with "Consequently, the presence of a greater number of
larger CCN can suppress..." or similar (be specific). Also replace
"prevents the smaller particles from activating" with "causing
fewer smaller particles to activate" or similar.

8) Introduction, page 2, lines 42-43 -- delete "the" between
"alter" and "precipitation" and change "rate" to "rates".

9) Introduction, page 2, line 44 -- avoid the term "direct effect"
here as that tends to be used in association with aerosol-radiation
interactions -- suggest to instead use "main effect" or other term.

10) Introduction, page 2, line 47 -- "an approximate -0.7 W/m2 increase"
change the phrasing to avoid negative increase -- also please give the
uncertainty range given by the radiative forcing chapter of the
IPCC report. You have "which is on the order of" but it's not clear
whether you are referring to the best-estimate value or the uncertainty
range itself. Please clarify.  Also use the recommended method to cite
chapters of the IPCC report -- you should cite the chapter authors rather
than the entire report -- i.e. Myhre et al. (2013) for the radiative
forcing chapter.

11) Introduction, page 2, line 52 -- You have "primary and secondary aerosol"
but I think you mean "primary and secondary particles" (i.e. you mean to
differentiate between particles that are directly emitted (primary) and
those which are generated separately via new particle formation.
Also suggest to replace "Primary particles are..."
with "For example, primary particles are...." at the sentence after.

12) Introduction, page 2, lines 55-56 --- whereas the preceding sentence
describes primary particles, here the text is explaining the production
of secondary aerosol *mass* rather than the presence of secondary particles
(i.e. those which are nucleated). The way this para is worded currently
could confuse the reader re: the distinction between secondary aerosol particles
and secondary aerosol mass. The sentences after can still be used in
relation to the secondary particles because the growth of the nucleated
(secondary) particles to climate-relevant sizes is strongly influenced
by the production of secondary organic aerosol (mass). So it's fine to
mention that the oxidation of VOCs to form SOA is an important
uncertainty, as you have done, but make it clear that the issue is how
the condensation of SOA causes growth of secondary particles (which
are only initially at nanometre sizes).

13) Introduction, page 3, lines 68-71 -- The first part of the
sentence beginning "It is therefore impractical..." (up to where
you have "...each compound individually") seems to be building up to
making a point about how models need to lump organic species together
into a small number of species representative of compounds with
similar properties.  Whereas the 2nd part (beginning "two popular
methods") then seems to refer to the separate issue of how to treat
the partitioning of the species to the particle phase.  I know
these issues are dependent on each other but in my mind they
are separate issues. I'd advise to finish off the sentence
finishing the point about the first part -- suggest to give a

reference to models that have lumped the organics in
different ways -- you could cite (for example) Oâ\200\231Donnell et al. (2011),
who have implemented a relatively complex representation of organics
compared to other global models -- suggest you also cite the
AeroCom organics intercomparison paper (Tsigaridis et al., 2014)
in terms of the typical (lack of) sophistication or organic aerosol
schemes in global aerosol models.

Then go on to make the points in that 2nd part with a separate sentence
or sentences explaining the issues around difficulties & advantages
of the different approaches to gas/particle partitioning.

14) Introduction, page 3, lines 74-75 -- re-word the excerpt of text
"The first noteworthy approach (Odum et al., 1996) proposes an..."
as "The first (Odum et al., 1996) involves an....". The "noteworthy
approach" seemed a little unscientific language.

15) Introduction, page 3, line 77 -- replace "these models are found to be..."
with "the approach has been found to be..." -- I think it's better
to refer to the approach used rather than a particular set of models.

16) Introduction, page 3, line 88 -- you have "particles above above 50%"
please correct -- do you mean "occurs only for relative humidity above 50%"?
Also -- at the end of the sentence you say "which is typically of atmospheric
relevance" -- re-word that -- what exactly is typically of atmospheric
relevance -- the rh value? Or do you mean this is always achieved everywhere
in the boundary layer or similar? Say why it's of relevance -- do you mean
that therefore this condition is always satisfied in most atmospheric conditions?

17) Introduction, page 3, lines 92-95 -- re-word this sentence -- you say the
existing particles are assumed to be involatile -- but is this a general
statement about what most models do? Or the model you are using -- please
clarify and improve the wording.

18) Introduction, page 3, lines 97-100 -- insert commas between "without it"
and "a polydisperse" -- and also between "particle case" and "the smaller..."

19) Introduction, page 4, lines 108-110 -- insert commas between "as a result"
and "the equilibrium" -- and between "organic mole fraction" and "a method...".

20) Section 2, page 4, lines 116-119 -- re-word this sentence that currently
begins "Multiple organic species" and delete the citation to Donahue et al. (2006)
in the first sentence (refer to it in the 2nd sentence when you actually
introduce the VBS approach)." I suggest to move the text "volatility basis set" to
the start of the sentence so that it begins something like "The volatility
basis approach (Donahue et al., 2006) involves binning multiple organic
species into a limited number of notional organic species with representative
saturation concentrations (volaility)...."  Just needs rewording.

21) Section 2, page 4, lines 119 -- delete the word "calligraphic".

22) Section 2, page 5, line 139 -- change notation to avoid use of the
term "C_OA" -- that notation is confusing beacuse the reader will expect
the OA to refer to organic aerosol whereas in fact it is representing
the total concentration of all compounds in the condensed phase.

23) Section 3, page 6, line 163 -- replace "size of the aerosol"
with "size of the aerosol particle" or just "size of the particle".

24) Section 4, page 7, line 194 -- replace "the only the molecules"
with "the only molecules".

25) Section 4, page 7, line 200 -- after "It is this quantity" suggest

to add in brackets exactly what you are referring to -- i.e. something
like "(the remaining concentration in the vapour phase)"

26) Section 4, page 7, caption to Figure 1 -- there seems to be a
word "equilibrate" on its own underneath the main caption -- presumably
this is a typo in the LaTeX or similar? Please correct.

27) Section 4, page 8, line 209 -- Re-word start of sentence beginning
"Dummy indices" -- they're not really dummy indices as they are
presumably chosen to provide a particular distinctive information
compared to the main indices, i and j. Suggest to re-word this as
"We choose to use a separate pair of indices r and k for summations, to
make clear..." -- and change "restricted" to "applied".

28) Section 4, page 9, lines 254-258 -- insert commas between "such problems"
and "but require" and between "of the paper" and "we present" -- also
replace "equations" with "equation" when referring to (24) (there's only 1).

29) Section 5, page 12, title of section 5.2 -- replace "with an Average
Mass Fraction" with "with a common (average) organic mass fraction".
Also -- it occurred to me here that I don't think you state that the
common organic mass fraction to be assumed is actually the average
value -- implicitly that's the case but I suggest you state that
up front -- and in the earlier instances where you mention this is one
of the key assumptions -- having the word "average" makes it seem a
bit more of a reasonable assumption to take I think.

30) Section 5, page 12, line 284 -- insert "organic" betwen "common"
and "mole fraction" -- perhaps also add "average" as well for the
same reason as above?

31) Section 5, page 12, line 309 -- suggest to replace "solved for"
with "solved by" and also replace "root find algorithm" with
"root-finder algorithm" or "root-finding algorithm".

32) Section 5, page 13, line 314 -- insert comma between "been removed"
and "it is sufficiently".

33) Section 5, page 13, line 320 -- insert comma between "parameter space"
and "values of the".

34) Section 5, page 13, caption to Figure 4 -- "are distinguished
by the shape of the points" -- say exactly what you mean here by
adding (circles are xx, squares are yy).

35) Section 5, page 13, caption to Figure 4 -- the sentence saying
"Equality of the solutions...." can be easily shortened substantially,
Just say that the dashed lines show the 2:1 lines.  Also it is very
difficult to distinguish between the dashed blue and the dashed green
(to my eye at least) -- suggest to make the 1:1 line a solid line and
the 2:1 and 1:2 lines dashed -- I think here the colours are not needed
(in fact they detract) and it's best just to use the black lines.

36) Section 5, page 13 -- Figure 4 -- there seems to be an error in
Figure 4 -- the dashed lines should be indicating the 2:1 line --
which they are at the top --- but at the bottom they are much narrower
than that -- presumably this is just a typo in the script used to
generate the graph right?  Please correct this. And add the legend
to the plot to make the Figure easier for the reader to understand.
This is also the case for Figure 6 on page 17.

37) Section 5, page 13 -- caption to Figure 4 -- please state in the
caption what the initial concentration of the sVOC is in these

integrations. Also add this to the caption to Figure 5.

38) Section 5, page 14 -- line 328 -- insert "organic" between
"common" and "mole fraction".

39) Section 5, page 14 -- line 343 -- replace "there increased
water contend" with "the increased water content".

40) Section 7, page 19 -- lines 417-418 -- the sentence beginning
"We derive a more accurate initial condition..."  This seems like
quite a strong limitation for a method intended to be generally
applicable for global models.  Please give a caveat here about
the fact that in a global model, this restriction would not be
possible.

41) Section 7, page 19 -- caption to Table 4 -- state what the
initial sVOC concentration is here (summed over the vapour and
particle phases).

42) section 7, page 20 -- the equation on line 461 has no number
-- please make sure it has a number (even though it may not be
referenced in the paper, that's still important).  Also please
change the lower-case "d" to upper-case "D" -- I asked this
already in my initial submission review to make this change.
Using lower case here is doubly confusing because the lower-case
d is used in the derivative term.  I suggest also to use upper
case N rather than lower case n as the usual way of presenting
a size distribution is "dN/dln D" (capital N and caption D).
Please make this change throughout.

43) section 7, page 21, lines 474-475 -- sentence beginning
"To speed up" --- this again seems rather a severe restriction
to make on the method --  see also point 40 -- there needs to
be some mention of how this restriction may limit the method's
applicability in a global model. How dependent is the method
on this and the restriction in point 40?

44) section 7, page 22, lines 497-500 -- "over predict" to
"over-predict" and "under estimate" to "underestimate".

45) section 7, page 22, line 500 -- the errors mentioned
here in the text sound potentially rather large.  Are the
authors saying the "cost" of these magnitude errors are a
good "investment" since they are small compared to the
"benefit" of being able to represent the partitioning of
the semi-volatile organics in a global model? Please add
a sentence clarifying this.

46) section 7.3.2 -- page 24 -- this section needs to be
improved considerably -- the text on line 570 is unclear
-- "two methods of approximating this size" -- what size?
You mean the overall size distribution? Also the end of
the sentence (line 571) says "that represent two regimes".
What are the two regimes here -- need to be much more
specific here about what you're referring to.

47) section 7.3.2 -- page 24, lines 574-576.  The text
"This method.... maintains a constant arithmetic standard
deviation and decreases the geometric standard deviation
in order to maintain mass within the system". I don't
understand this method at all -- particle size distributions
tend to follow lognormal size distributions which means
that the geometric standard deviation is the appropriate

measure for the width of the modes. In any case the two
measures (geometric stdev and arithmetic stdev) are
related and if one is kept constant the other should stay
constant too.  Or am I missing something here? There needs
to be better explanation of this method or the reader is
left puzzled.

48) Section 7.3.2 -- page 24, line 590 -- What is the change
in conditions that is being simulated here? What are the 2 hour
and 24 hours indicating a change from -- the experiments
being carried out are not sufficiently described here -- this
needs to be further clarified before the paper can proceed to
publication.

49) Section 7.3.2 -- page 24, lines 592-593 -- "This is in
agreement with the initial proposal put forward by Connolly
et al. (2014)" -- need to state explicitly what part of the
proposal you're referring to here -- what is the point you
are making here -- extend the length of the sentence to allow
the issue to be properly communicated to the reader.

50) Conclusions, page 25 -- there needs to be included much
more prominently in the conclusions mention of the
simplifications applied in the method -- in particular the
techniques used in the box model re: the initialisation of
the simulations. It simply would not be possible to do that
in a global model and I am not yet convinced therefore that
the applicability of the method is as straightforward as
the authors are making out.

51) Conclusions, page 25, line 625 -- add comma after
"of the involatile compounds".

52) Page 35, caption to Figure 9 -- change "pcc" to "cm$^{-3}$"
(assuming in LaTeX form). Also in other Figures.

53) Page 35, caption to Figure 9 -- need to add somewhere in
the caption that the 2nd mode has diameter at 125nm and its
number concentration given -- the reader needs that info to
be at hand (in the caption) in order to be able to effectively
interpret what is shown. Also needs to be stated in Figures 10 and 11
and in Figures 14, 15, 16 and 18.

54) Page 36, caption to Figure 10 -- "models" --> "modes".

55) Page 46, caption to Figure 20 -- state what the initial conditions
are here that are being integrated.

References

Myhre, G., and Coauthors, 2013: Anthropogenic and natural radiative
forcing. Climate Change 2013: The Physical Science
Basis, T. F. Stocker et al., Eds., Cambridge University Press,
659â\200\223740

Oâ\200\231Donnell, D., Tsigaridis, K., and Feichter, J.: Estimating the
direct and indirect effects of secondary organic aerosols using
ECHAM5-HAM, Atmos. Chem. Phys., 11, 8635â\200\2238659,
doi:10.5194/acp-11-8635-2011, 2011

Tsigaridis, K. Daskalakis, N., Kanakidou, M. et al.,
Atmos. Chem. Phys., 14, 10845â\200\22310895, 2014.